# Molecular evidence and ecological niche modeling reveal an extensive hybrid zone among three *Bursera* species (section *Bullockia*)

**Eduardo Quintero Melecio**[1☯], **Yessica Rico**[1☯*], **Andrés Lira Noriega**[2,3],
**Antonio González Rodríguez**[4]

**1** Red de Diversidad Biológica del Occidente Mexicano, Centro Regional del Bajío, Instituto de Ecología, A. C., Pátzcuaro, Michoacán, Mexico, **2** Red de Estudios Moleculares Avanzados, Instituto de Ecología, A.C., Xalapa, Veracruz, Mexico, **3** CONACyT, Ciudad de México, Mexico, **4** Laboratorio de Genética de la Conservación, Instituto de Investigaciones en Ecosistemas, Universidad Nacional Autónoma de México, Morelia, Michoacán, Mexico

☯ These authors contributed equally to this work.
* yessica.rico@inecol.mx

**Data Availability Statement:** The sequences analyzed in this study are available in NCBI GENBANK (www.ncbi.nlm.nih.gov) accession numbers: ETS: B. cuneata (OK413888 -

## Abstract

The genus *Bursera*, includes ~100 shrub and trees species in tropical dry forests with its center of diversification and endemism in Mexico. Morphologically intermediate individuals have commonly been observed in Mexican *Bursera* in areas where closely related species coexist. These individuals are assumed to result from interspecific hybridization, but no molecular evidence has supported their hybrid origins. This study aimed to investigate the existence of interspecific hybridization among three Mexican *Bursera* species (*Bullockia* section: *B. cuneata*, *B. palmeri* and *B. bipinnata*) from nine populations based on DNA sequences (three nuclear and four chloroplast regions) and ecological niche modeling for three past and two future scenario projections. Results from the only two polymorphic nuclear regions (PEPC, ETS) supported the hybrid origin of morphologically intermediate individuals and revealed that *B. cuneata* and *B. bipinnata* are the parental species that are genetically closer to the putative hybrids. Ecological niche modeling accurately predicted the occurrence of putative hybrid populations and showed a potential hybrid zone extending in a larger area (74,000 km$^2$) than previously thought. Paleo-reconstructions showed a potential hybrid zone existing from the Last Glacial Maximum (~ 21 kya) that has increased since the late Holocene to the present. Future ecological niche projections show an increment of suitability of the potential hybrid zone for 2050 and 2070 relative to the present. Hybrid zone changes responded mostly to an increase in elevational ranges. Our study provides the first insight of an extensive hybrid zone among three Mexican *Bursera* species based on molecular data and ecological niche modeling.

OK413898), B. palmeri (OK413914 - OK413917),
B. bipinnata (OK413900 - OK413913); (2) PEPC: B.
cuneata (OK413918 - OK413925), B. palmeri
(OK413927 - OK413930), B. bipinnata (OK413931
- OK413939).

**Funding:** This study was funded by Consejo
Nacional de Ciencia y Tecnología (CONACYT www.
conacyt.gob.mx) through the Ciencia Básica
CB2016-283237 project to YR and a Master's
scholarship to EQM (MM18019). The funders had
no role in study design, data collection and
analysis, decision to publish, or preparation of the
manuscript.

**Competing interests:** The authors have declared
that no competing interests exist.

## Introduction

Interspecific gene flow resulting in the formation of intermediate individuals across hybrid zones is a common phenomenon in nature with important consequences for evolution and conservation [1–3]. Hybridization can reinforce the development of reproductive barriers, introduce potentially adaptive genetic variation into a population driving ecological divergence, and lead to the formation of new lineages [4–6]. Alternately, hybridization can result in maladaptive genetic introgression, displacement of pure parental species by novel phenotypes, formation of highly invasive lineages, and species extinction through genetic assimilation [7–9]. The study of hybrid zones has been a prolific area of research since it is key for understanding the factors that promote reproductive isolation, ecological adaptation, and speciation in natural and human-modified habitats [3, 10, 11]. Hybridization is common in plants [12] but there are still many plant taxa in which hybridization is poorly understood [13].

The genus *Bursera* encompasses a diverse group of nearly 100 species of deciduous and resinous shrub and trees occurring from the south-western USA to Peru, the Galapagos, Bahamas, and the Greater Antilles. The genus comprises two monophyletic sections, *Bursera* and *Bullockia*, which diversified in the early Miocene with the increase of aridification in Mesoamerica and expansion of the Tropical Dry Forest (TDF) [14]. The highest species diversity is found in the Pacific drainages of western Mexico, and nearly 90% of the species are endemic to the country [15, 16]. *Bursera* species have economic and cultural importance as they are used for the extraction of aromatic resins for religious and medicinal purposes [17], for elaboration of wood handcrafts, and for *ex-situ* propagation in restoration programs or as living fences in vast extensions throughout tropical landscapes [18].

Several taxonomic studies have documented the occurrence of morphologically intermediate individuals in areas where closely related *Bursera* species co-occur, likely as the result of secondary contact [16, 19–23]. Interspecific hybridization and introgression have been inferred as important drivers of speciation in several *Bursera* species [16], although only few species have been recognized to have a hybrid origin based on molecular and biochemical evidence [21, 24]. The most relevant study of interspecific hybridization was performed by Weeks and Simpson [21] who confirmed the hybrid origin of three endemic *Bursera* species (*B. brunea*, *B. gracilipes*, and *B. ovata*) from Hispaniola in the Caribbean based on nuclear and chloroplast DNA sequences. However, a subsequent study by Weeks and Tye [25] did not find evidence for a hybrid zone in the Galapagos Islands between *B. malacophylla* and *B. graveolens*, but instead suggested that *B. malacophylla* and the intermediate individuals shared ancestry of *B. graveolens*, a species of high morphological variation across the islands. Weeks and Tye [25] implied that the presence of morphologically intermediate individuals in areas of species co-occurrence cannot be taken as unambiguous evidence of interspecific hybridization as many *Bursera* species exhibit morphological plasticity across their ranges [16, 26].

Besides the use of molecular data in studies of interspecific hybridization, recent approaches have incorporated ecological niche modeling for inferring the geographic occurrence of potential hybrid zones and the role of ecological factors influencing the development of ecological divergence and local adaptation [5, 27–29]. In this context, past and future climatic projections of hybrid zones can provide a complementary perspective for understanding the origin, maintenance, and temporal dynamics of hybrid zones in association with global climatic changes [30, 31].

*Bursera* species are dominant or codominant woody elements of the TDF, thornscrub, and desertscrub of Mexico [32, 33], and some species can also occur in secondary habitats of oak and pine forests [19]. The TDF of the Bajío in Central-West Mexico was one of the most widespread ecosystems in the region, which is currently disappearing due to anthropogenic land

use changes [34]. In this region, between the states of Michoacán and Guanajuato, Rzedowski and Guevara-Féfer [19] reported the ocurrence of putative *Bursera* hybrids with morphologically intermediate leaf characteristics between *B. bipinnata*, *B. cuneata*, and *B. palmeri*. These *Bursera* species are mostly allopatric with overlapping ranges in the northeastern Michoacán. Despite the observations that hybridization is considered frequent in Mexican *Bursera* [16], there is no molecular evidence confirming this phenomenon, while no information exists on the presence of hybrid zones between co-occurring *Bursera* species.

In this study we examined putative hybrid populations using nuclear and chloroplast DNA sequences coupled with ecological niche modeling to ask the following questions: (*i*) Are morphologically intermediate individuals of hybrid origin from *B. cuneata*, *B. palmeri* and *B. bipinnata*? (*ii*) To which parental *Bursera* species do putative hybrids show the closest genetic affinity? (*iii*) Do the putative hybrids occur in a similar ecological niche to the parental species? (*iv*) Can we spatially and environmentally predict the formation of the hybrid zone through ecological niche modeling? (*v*) What is the role of past and future climatic conditions on the formation and maintenance of this hybrid zone?

## Material and methods

### Target species and sampling

*Bursera palmeri*, *B. cuneata* and *B. bipinnata* are dioecious and deciduous trees in the *Bullockia* section which is characterized by having well-developed cataphylls, bilocular ovaries, fruits with two valves, and smooth, grey, not exfoliating barks [16]. There is no information on the chromosome number in these species, but they are expected to be diploid as in other *Bursera* [35].

Leaf shape can differentiate the putative hybrids from their potential parental species: *B. bipinnata* has small bipinnate leaves, *B. cuneata* has larger, oblong to lanceolate, once-pinnate leaves of leathery texture and rugose appearance in the upper side. *Bursera palmeri* and *B. cuneata* are morphologically more similar, but *B. palmeri* has leaves with membranous texture and leaflets not rough on the upper side. The putative hybrids exhibit intermediate leaf characteristics including partial bipinnate leaves with large variation in size, texture, and level of leaf divisions (Fig 1). The bipinnate characteristic suggests that *B. bipinnata* is one of the parental species. We have not identified intermediate individuals between *B. cuneata* and *B. palmeri*.

According to Rzedowski and Guevara-Féfer [19] and herbaria specimens deposited in the IEB herbarium (Instituto de Ecología A.C., Mexico) we identified various putative hybrid populations. During May 2017 to November 2019, we randomly collected fresh leaf samples from adult or juvenile trees of the three *Bursera* species and their putative hybrids in nine localities: four representing hybrid populations as identified by the co-occurrence of putative hybrid trees and their potential parental species and four populations of the parental species where neither of the other two *Bursera* species were present, thus representing "pure" parental populations (Table 1; Fig 1). Leaves were preserved in sealable plastic bags containing silica gel until DNA extractions were performed. GPS coordinates were recorded for each locality.

### Laboratory procedures

Genomic DNA from 20 mg of dried tissue was extracted following the CTAB extraction protocol of Doyle and Doyle [36]. We amplified four chloroplast (cpDNA) intergenic spacers (*psbA-trnH*, *rbcL*, *trnK-matK*, *rps16*) and three nuclear regions including the external transcribed spacer ETS, fourth intron of the phosphoenolpyruvate carboxylase PEPC, and third intron of the NIA-i3. These regions have been used in phylogenetic and hybridization studies of *Bursera* (e.g., [14, 21]). PCR reactions were performed using the MyTaq™ DNA polymerase

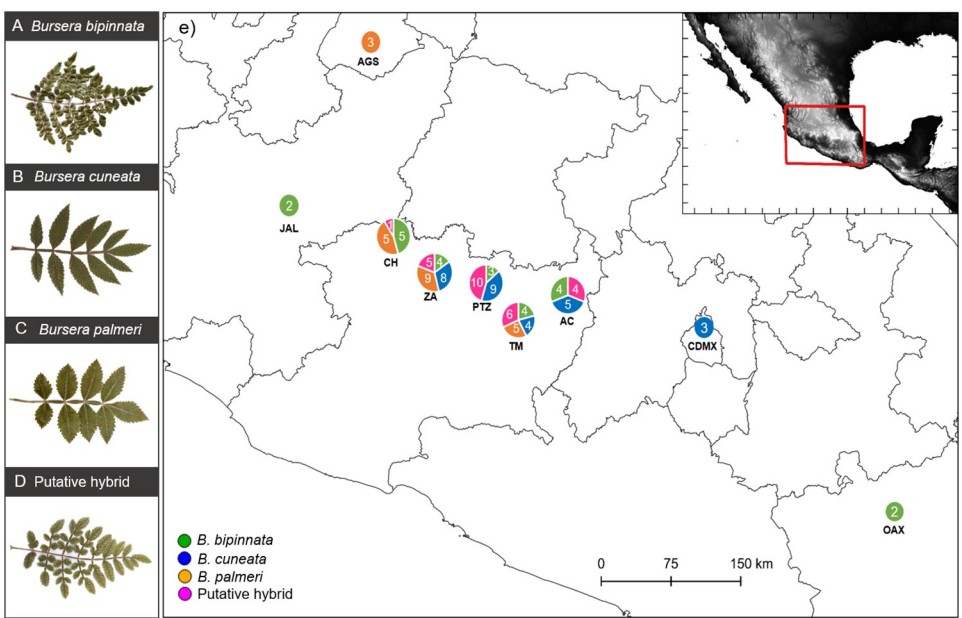

**Fig 1. Leaf characteristics of (A) *Bursera bipinnata*, (B) *B. cuneata*, (C) *B. palmeri* and (D) putative hybrids, and (E) map of sampling localities.** Numbers within circles represent the number of samples collected per each species and the putative hybrids per locality; color designations: *B. bipinnata* (green), *B. cuneata* (blue), *B. palmeri* (yellow), and putative hybrids (pink). Sampling localities: Aguascalientes AGS, Jalisco JAL, Churintzio CH, Zacapu ZA, Pátzcuaro PTZ, Tarímbaro TM, Acumbaro AC, Ciudad de Mexico CDMX, Oaxaca, OAX.

kit (BIOLINE, London, United Kingdom) as follows: 5 μL of PCR buffer, 0.4 μL (10 μM) of each forward and reverse primer, 0.2 μL (5 U/ μL) of Taq polymerase and 5–10 ng of genomic DNA in a final volume of 25 μL. PCR conditions were as follows: 5 min denaturation at 95 ˚C, 35 cycles of 60 s denaturation at 95 ˚C, 60 s annealing at 53–56 ˚C, 90 s extension at 72 ˚C, and a final extension of 10 min at 72 ˚C. PCR products were run on a 1% agarose gel, stained with GelRed™ (BIOTIUM, Fremont, California) and visualized under UV light. Unpurified, amplified PCR products were sent for sequencing to MAGROGEN Inc. (Seoul, South Korea). We included a positive and negative control to check for contamination. Quality of chromatogram files were checked and edited using CHROMAS v2.6.5 (Technelysium Pty Ltd); unambiguous nucleotide sites for the nuclear regions were coded using the IUPAC nomenclature. Two

**Table 1. Sampling localities for three *Bursera* species and their putative hybrids in Mexico.**

| Locality | Code | State | Status | Latitude | Longitude | Species ID |
|---|---|---|---|---|---|---|
| Churintzio | CH | Michoacán | Hybrid | 20.14595 | -102.02563 | Bm/Bp/Bh |
| Zacapu | ZA | Michoacán | Hybrid | 19.92654 | -101.74077 | Bc/Bm/Bp/Bh |
| Lago Pátzcuaro | PTZ | Michoacán | Hybrid | 19.58867 | -101.62238 | Bc/Bp/Bh |
| Tarímbaro | TM | Michoacán | Hybrid | 19.82512 | -101.21947 | Bc/Bm/Bp/Bh |
| Acumbaro | AC | Michoacán | Hybrid | 19.62107 | -100.46406 | Bc/Bp/Bh |
| Ciudad de México | CDMX | Capital City | Parental | 19.56973 | -99.12353 | Bc |
| Calvillo | AGS | Aguascalientes | Parental | 21.90872 | -102.63875 | Bm |
| Chapala | JAL | Jalisco | Parental | 20.29604 | -103.20277 | Bp |
| San Miguel Etla | OAX | Oaxaca | Parental | 17.21702 | -96.75739 | Bp |

Species abbreviations: *B. bipinnata* Bp, *B. cuneata* Bc, *B. palmeri* Bm, putative hybrids Bh. Sample sizes for each locality see Fig 1.

cpDNA regions (*psbA-trnH*, *rbcL*) and *NIA-i3* consistently failed to amplify, while the other two cpDNA regions (*trnK-matK*, *rps16*) resulted non-polymorphic in a subset of DNA samples for each species (GENBANK accessions: *trnK-matK*: OK413940—OK413942, *rps16*: OK438213—OK438215). Thus, our results are based on the polymorphism of the ETS and PEPC genes. We successfully sequenced 101 individuals, and these sequences are available in NCBI GENBANK ([www.ncbi.nlm.nih.gov](www.ncbi.nlm.nih.gov)) with accession numbers: (1) ETS: *B. cuneata* (OK413888—OK413898), *B. palmeri* (OK413914—OK413917), *B. bipinnata* (OK413900—OK413913); (2) PEPC: *B. cuneata* (OK413918—OK413925), *B. palmeri* (OK413927—OK413930), *B. bipinnata* (OK413931—OK413939).

## Recombination, phylogenetic relationships, and haplotype networks

Sequences were aligned using MUSCLE v3.8.31 [37] with default parameters and manually adjusted in MEGA X [38]. The gametic phase of heterozygous individuals was determine using PHASE 2.1 [39] as implemented in DNASP v6.12.03 [40]. The algorithm was run with 1000 iterations, 20% burn-in, a thinning interval of 10, and 0.9 as the minimum posterior probability of haplotypes. We performed all subsequent analysis with the phased sequences.

Evidence of recombination, which would be expected from interspecific hybridization, was evaluated with the pairwise homoplasy index (PHI-test) as implemented in SplitsTree v4.17.1 [41]. Moreover, we ran seven different detection recombination methods (RDP, GENECONV, BootScan, MaxChi, Chimaera, SiSan, and 3Seq) in the software RDP v4.101 [42]. We used the default parameters and considered the recombination events that were supported for three out of the seven selected methods.

We reconstructed phylogenetic relationships with the concatenated matrix using a Bayesian (BI) method. We used as outgroup sequences of other *Bursera* species retrieved from NCBI GenBank: *B. simaruba* (accession number: AF445946, GQ377992), *B. kerberi* (accession number: JF919125, AF445937) and *B. fagaroides* (accession number: AY309365, AY964604). We also sequenced a specimen of *B. penicillata* from Chapala Jalisco (Section *Bullockia*; accession number: OK413926, OK413899). The best model of sequence evolution was selected using the Akaike information criterion (AIC) in JMODELTEST v2.1.10 [43]. BI analyses were carried out using BEAST v2.5 [44]. We used BEAUti v1.8.1 [45] to adjust a substitution rate GTR gamma with invariant sites, a log normal relaxed molecular clock model to estimate the substitution rate. The analysis was run for 100 million generations for three independent runs. Posterior values were sampled every 1,000 generations and a burn-in of 10%. The BEAST output file was evaluated using TRACER v1.7 [46]. The less likely trees (probability > 0.8) were discarded using TreeAnnotator v1.8.2 [47]. The consensus tree was visualized with FigTree v1.4.2 ([http://tree.bio.ed.ac.uk/software/figtree/](http://tree.bio.ed.ac.uk/software/figtree/)).

We then constructed an unrooted phylogenetic network with the neighbor-net algorithm implemented in SplitsTree v4.17.1 [41] using a concatenated matrix of unique haplotypes for each species and putative hybrids. The neighbor-net algorithm is optimal for visualizing reticulated relationships among sequences, as it is expected from interspecific hybridization [41]. Additionally, haplotype relationships were observed with the median-joining (MJ) algorithm, and excluding gaps, as implemented in POPART [48]. We constructed a MJ network for each marker separately to visualize if the most frequent haplotypes for each species were shared with the putative hybrids irrespective of the nuclear marker.

## Genetic diversity and differentiation

For each species and the putative hybrids, we estimated the number of haplotyes (*h*), haplotype diversity (*Hd*), nucleotide diversity (π), and expected heterozygosity (*Het*) using ARLEQUIN

v3.5.2.2 [49]. We used BAPS v5.3 [50] to identify the most likely number of genetic clusters and genetic assignments between the potential parental species and their putative hybrids. We used the clustering of linked molecular data option, which is appropriate for diploid sequence data. First, we performed few exploratory runs from $K$ = 2 to 15 and few replicates for each $K$ (n = 5) to determine the most likely genetic clusters. After these initial searches, we found a likely number between 5 and 7. Thus, we carried out three independent runs from $K$ = 3 to 9 with 10 replicates each. The most likely number of $K$ clusters was determined by its higher posterior probability and likelihood value. Then, we used the binary output file of the mixture clustering to run an admixture analysis. We used 1000 iterations, 5 reference individuals for each population and 100 iterations for the reference individuals. These parameters were defined by performing several exploratory runs and checking the consistency among runs. To assess patterns of genetic differentiation among the three *Bursera* species and the putative hybrids, we carried out a Discriminant Analysis of Principal Components (DAPC) in the R library *adegenet* [51] and used a binary coded matrix of presence (1) and absence (0) of haplotypes for each marker per individual. The DAPC is a multivariate method free of Hardy-Weinberg and linkage disequilibrium assumptions [51]. We used the three species and the putative hybrids *a priori* within the dapc function, while the optimal number of PCs to retain was optimized using the function xvalDapc [51]. Pairwise $F_{ST}$ comparisons between the four groups were calculated with 1000 permutations to test for statistically significant differences in ARLE-QUIN v3.5.2.2 [49].

## Ecological niche modeling and niche overlap zone

We compiled herbarium records for the three *Bursera* species and the putative hybrids from different web databases, including the Global Biodiversity Information Facility (GBIF; http://www.gbif.org/), SEINet (https://swbiodiversity.org/seinet/), Missouri Botanical Garden (http://www.tropicos.org), National Herbarium of Mexico (MEXU; http://www.ib.unam.mx/botanica/herbario/), and the IEB Herbarium in Mexico. In addition, we requested *Bursera* herbarium records, including putative hybrids, from 33 Mexican universities and institutional herbaria (S1 Table). Specimen photography, collector and identifier were consulted to corroborate the veracity of specimens. Our search retrieved 1,529 records, which included the entire distribution range of the three parental *Bursera* species and 19 records of putative hybrids. To reduce sampling biases for the niche modeling, the occurrences were spatially thinned at 20 km for *B. bipinnata* and *B. palmeri*, and 15 km for *B. cuneata*. The variable radii were determined based on the number of available records and the spatial extent of each species distribution. After this filtering, the dataset comprised 313 occurrences for *B. bipinnata*, 105 for *B. palmeri*, and 93 for *B. cuneata*. Each species occurrences were equally split in two datasets, one for building the niche model and the second for data testing. No spatial thinning was applied for the putative hybrid records. These records were used only to carry out the similarity and equivalency test (see below).

We selected 11 bioclimatic layers in 30 arc-seconds from WorldClim (https://worldclim.org/data/worldclim21.html) [52]: Annual mean temperature (BIO1), mean diurnal range (BIO2), isothermality (BIO3), temperature seasonality (BIO4), temperature annual range (BIO7), annual precipitation (BIO12), precipitation of the wettest month (BIO13), precipitation of the driest month (BIO14), precipitation seasonality (BIO15), precipitation of the warmest quarter (BIO18), precipitation the coldest quarter (BIO19). This layer selection was based on other niche model and distributional studies in *Bursera* species [53, 54]. Also, *Bursera* occurrence at finer spatial scales is determined by slope aspect and roughness of the ground [55], thus we generated a topographic layer based on a SRTM digital elevation model (https://

www2.jpl.nasa.gov/srtm/). The calibration areas were delimited using the WWF biogeographic regions to delimit the species accessible areas [56]. We discarded highly correlated variables ($\geq 0.8$) using Spearman´s rank correlation coefficients and removed variables with the lowest contribution in the final model measured by Jackknife analysis using the R library *raster* [57].

Niche models under current climatic conditions using the above-mentioned variables were constructed using the maximum entropy algorithm implemented in MAXENT v3.4.1 [58] using the R library *kuenm* v1.1.1 [59]. Levels of model complexity were evaluated, such as over-fitting by varying the regularization multiplier (RM) (0.1, 0.5, 1, 2, 3, 4) and feature classes linear (L), quadratic (Q), product (P), threshold (T), and hinge (H) in 5 combinations (i.e., L, LQ, LQP, LQPT, LQPTH), and 10000 background points for *B. bipinnata* and *B. palmeri* and 7000 points for *B. cuneata* due to its smaller range. This resulted in 150 candidate niche models for each parental species. The niche models were evaluated based on the statistical significance of the partial Receiver Operating Characteristic (pROC) [60], 10000 random points, 500 iterations, and 0.5% omission rates. We generated several metrics to rank the best models: AUC, omission rate, AICc, delta AICc, parameter number, and WAICc. The niche model selected with the best parameters was constructed and projected in the same polygon with ten bootstrap iterations. The best niche model for each *Bursera* species were set to thresholds with the 10 percentile training presence values to produce binary raster maps. These binary maps were overlapped and the areas of conjunction of pixels among the three parental species were defined as the niche overlap zone. This procedure was carried out using QGIS v2.12 [61]. The 19 putative hybrid records were mapped on the overlapping niche zone to determine their spatial correspondence.

## Niche equivalency and similarity tests

To test for ecological niche divergence as evidence for reproductive isolation [62] among parental *Bursera* species and putative hybrids, we used the approach based on principal component analyses of climatic and topographic variables (PCA-env) to perform multivariate comparisons of niche overlap between pairs of species and the putative hybrids [63]. PCA-env uses a kernel density function to compute the density of occurrences in the multivariate PCA space, to account for potential bias from unequal sampling effort. This procedure consists in three steps: (*i*) calculation of the density of occurrences and environmental factors across the axes of the PCA-env, (*ii*) evaluation of the niche superposition along the gradient of multivariate analysis, and (*iii*) comparisons with repetitions of randomly generated simulated values of the Schoener's D metric [64] and the modified Hellinger distance I [65]. Based on ecological niche modeling of *Bursera* species, we performed the PCA with the most common and important variables according to the permutation importance. We generated four environmental axes (PCA-env) and selected the two with the highest variance explained. With the selected two PCA-env axes, we extracted the environmental values of the occurrence points and the calibration areas for each species, the putative hybrids, and for the total occurrence points, i.e., the four groups, in the study area. The overlap niche zone was used as the hybrid calibration area. With this, we constructed the species object of each parental *Bursera* species and the putative hybrids, which represents the environmental tolerance threshold of each species. The species objects were used to compute the D and I metrics, which calculate the niche overlap ranging from 0 (no overlap) to 1 (complete overlap). In this case, the niche overlap was calculated between parental species and between a parental species and the putative hybrids. Then, using the two metrics, we performed niche equivalency and similarity tests that compare the niche overlap values to a null distribution. The equivalency test evaluates whether two niches are significantly different from each other (not environmentally identical), while the niche

similarity test evaluates whether two niches are more similar (niche A can predict niche B and vice versa) than expected by chance [66]. We used the option "alternative" to test for niche conservatism, i.e., greater, in the R library *ecospat* [63] by performing 5,000 permutations and 100 replicates.

## Paleodistributions and future climatic projections

To reconstruct the historical dynamics of the potential hybrid zone, i.e., overlap niche zone, we projected the ecological niche model under current climatic conditions into three past scenarios: Last Interglacial (LIG ~ 120–140,000 ya) at 30 seconds [67] downloaded from World-Clim (http://www.worldclim.com), Last Glacial Maximum (LGM ~ 21,000 ya) and late Holocene (LH ~ 2,200 ya) at 2.5 min resolution and using the global models MIROC-ESM (Model for Interdisciplinary Research on Climate) [68] and CCSM4 (The Community Climate System Model, https://www.cesm.ucar.edu/models/ccsm4.0/) [69] obtained from PaleoClim (http://www.paleoclim.org/). To project future climatic scenarios of the hybrid zone, we projected niche models at the years 2050 and 2070. In both cases we employed the global models CCSM 4.0 and the MIROC5 based on two scenarios of Representative Concentration Pathway (RCP) referred as RCP4.5 (optimistic) and RCP8.5 (pessimistic) [70]. The RCP 8.5 is the scenario with the highest predicted greenhouse gas emissions compared with RCP 4.5 [71, 72]. All projections were set to thresholds with 10 percentile training presence values to produce raster binary maps. The binary maps of each species were overlapped for each time scenario and the areas of conjunction of pixels among the three parental species were defined as the niche overlap zone for each time. This procedure was performed in QGIS v2.12 [61]. Lastly, to assess range shifts in the niche overlap zone in response to the climatic changes, we extracted elevation values from 500 random points from the binary maps of the different time niche scenarios for the CCSM4 and MIROC5. We constructed density plots with the 500 random points to observe the altitudinal range differences between scenarios for the overlap niche zone.

## Results

### Recombination, phylogenetic and haplotype relationships

The concatenated ETS (421 bp) and PEPC (299 bp) aligned matrix included 202 phased sequences and had a total of sites excluding gaps or missing data of 720 bp with 38 parsimony informative sites. The PHI-test of recombination was statistically significant ($P < 0.027$). RDP found evidence of two recombination events, but only one of them was supported for three different methods (MaxChi, SiScan, 3Seq, start-end position 25–328, $P = 0.001$). This recombination event involved an individual from *B. cuneata*, whose major parent was a putative hybrid, and a minor parent was an individual of *B. cuneata*. These three individuals were from Acumbaro, Michoacán (AC).

The best-fit model of sequence evolution was HKY gamma with invariable sites for the ETS and GTR gamma with invariant sites for the PEPC. The BI phylogenetic tree resulted in a high-supported *Bullockia* clade (posterior probability = 1), but the three parental species were not monophyletic. Putative hybrid sequences were intermixed among subclades of the three parental species in high proportion and as sister subclades of *B. cuneata* and *B. bipinnata* (S1 Fig). Results from the neighbor-net in SplitsTree showed a highly reticulated tree, where putative hybrid and parental haplotypes were intermixed among major clades. Like the BI tree, the three *Bursera* species were not monophyletic, of which *Bursera cuneata* haplotypes were the most intermixed with putative hybrids (Fig 2A).

The MJ haplotype network for the ETS gene showed 43 haplotypes, seven exclusive to *B. cuneata*, three to *B. palmeri*, nine to *B. bipinnata* and 16 to the putative hybrids. The three

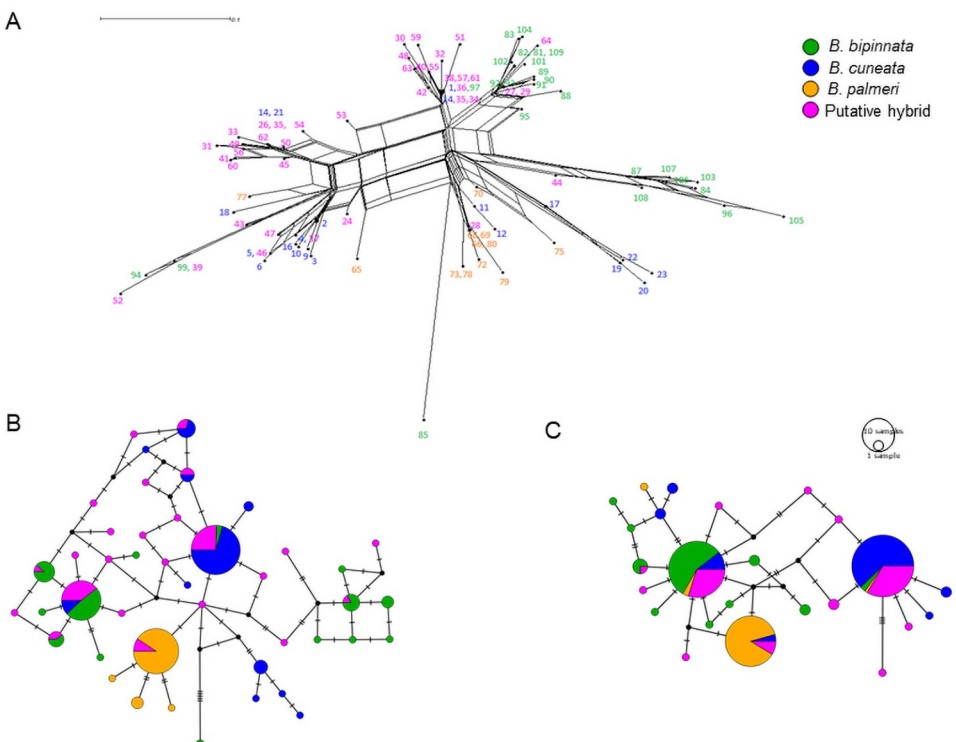

**Fig 2. Neighbor-net tree (A) and median-joining haplotype networks for 202 phased sequences of the (B) ETS and (C) PEPC nuclear genes in *Bursera bipinnata* (green), *B. cuneata* (blue), *B. palmeri* (yellow), and putative hybrids (pink).** Size of nodes is proportional to the number of sequences per haplotype, dashed lines represent the number of mutational steps separating haplotypes, and black nodes represent unsampled haplotypes.

most frequent haplotypes in each of the three *Bursera* species were shared with the putative hybrids. For *B. palmeri* none of the haplotypes were shared with any of the other two parental species, while *B. cuneata* and *B. bipinnata* shared two of the most frequent haplotypes (Fig 2B). The MJ haplotype network for the PEPC gene showed 26 haplotypes, four exclusive to *B. cuneata*, seven to *B. bipinnata*, one to *B. palmeri*, and eight to the putative hybrids. A large proportion of the putative hybrid sequences were shared among the most common haplotypes of the three *Bursera* species. These common haplotypes were shared among the parental species, although in lower proportions relative to the putative hybrids (Fig 2C).

**Genetic diversity and differentiation.** Estimates of genetic diversity showed moderate to high diversity values (Table 2). The putative hybrids exhibited the highest genetic diversity, including expected heterozygosity, closely followed by *B. bipinnata*. Of the three *Bursera*

**Table 2. Genetic diversity indices of the concatenated ETS and PEPC nuclear genes in three *Bursera* species and their putative hybrids.**

| Group | ID | n | h | π | Hd | Het |
|---|---|---|---|---|---|---|
| *B. bipinnata* | Bp | 46 | 23 | 0.009 | 0.913 | 0.197 |
| *B. cuneata* | Bc | 58 | 18 | 0.0059 | 0.756 | 0.175 |
| *B. palmeri* | Bm | 44 | 7 | 0.0014 | 0.332 | 0.072 |
| Putative hybrids | Bh | 54 | 32 | 0.009 | 0.948 | 0.258 |

Number of phased sequences (n), number of haplotypes (*h*), nucleotide diversity (π), haplotype diversity (*Hd*), and expected heterozygosity (*Het*).

species, *B. palmeri* showed the lowest diversity values, which haplotype diversity was four-fold lower than the putative hybrids.

Results from BAPS showed that the most likely number of genetic clusters was $K = 7$ (log marginal likelihood = -2633.06, PP = 0.99). Admixture analysis showed that the putative hybrids showed a high proportion of individuals assigned to each of the three parental *Bursera* species, but also individuals of each *Bursera* species were assigned to the putative hybrids, of which *B. bipinnata* showed the larger number of individuals assigned to the putative hybrids (Fig 3A). Few individuals showed admixed ancestry between two or three clusters, but all of them shared ancestry with the putative hybrids (Fig 3A). The DAPC plot showed the putative hybrids in-between individuals of the three parental species, but with closer affinity to *B. cuneata* and to a lesser extent to *B. palmeri*. The DA 1 axis showed the separation of *B. palmeri* from the rest, while DA 2 axis showed a larger separation of *B. bipinnata*, although there was a considerable overlap with the putative hybrids (Fig 3B). Pairwise $F_{ST}$ genetic distances showed that *B. cuneata* and the putative hybrids were genetically closer relative to the other two species, while *B. palmeri* was the most distant to the putative hybrids (S2 Table).

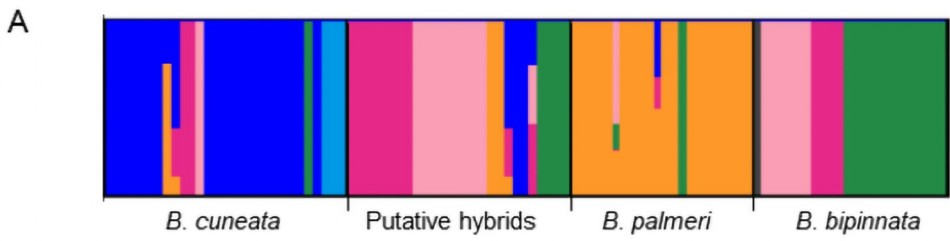

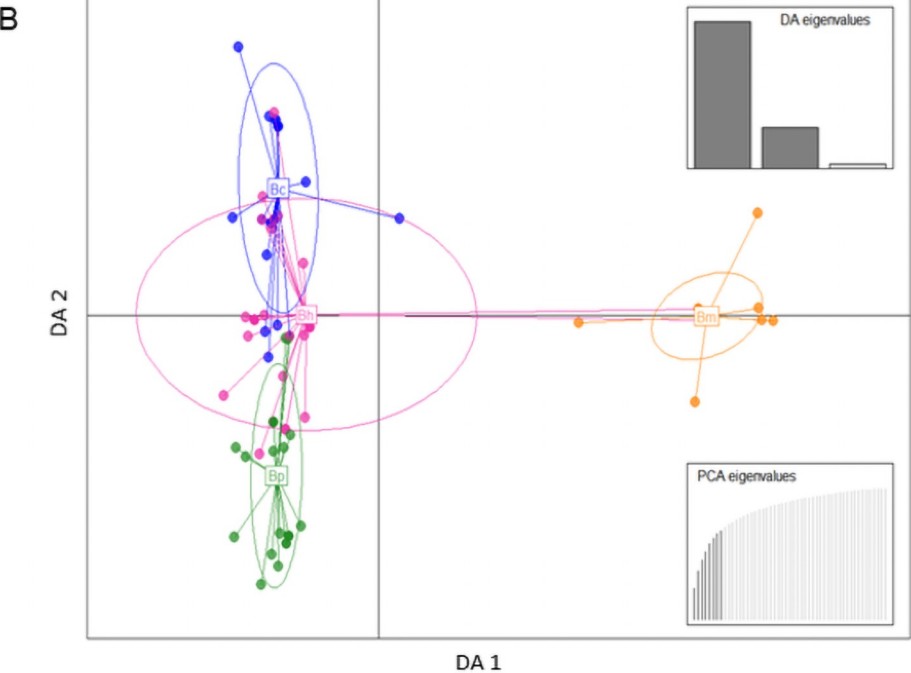

**Fig 3. (A) BAPS barplot of the best-supported *K* = 7 clusters and (B) DAPC plot of haplotypes in *Bursera bipinnata* (Bp), *B. cuneata* (Bc), *B. palmeri* (Bm), and the putative hybrids (Bh).** The insets showed the proportion of variance explained by each of the two DA eigenvalues (DA 1 & 2 respectively) and the number of retained PCA eingenvalues.

## Ecological niche modeling and overlap niche analyses

According to the permutation values, the most important variables among the three *Bursera* species were temperature seasonality (BIO4), precipitation seasonality (BIO15), annual mean temperature (BIO1), and precipitation of the wettest quarter (BIO13). The AUC values for the niche models were high for the three species: *B. bipinnata*, 0.914 ± 0.03, *B. palmeri*, 0.938 ± 0.007 and *B. cuneata*, 0.900 ± 0.009, indicating that occurrence points were strongly differentiated from background locations, so model distributions were not random.

The *B. bipinnata* ecological niche model exhibited high suitability along the mountains of the Pacific coast from Nayarit up to Guatemala, and largely comprising the Bajío and the Balsas regions, and with altitudinal range between 900 to 1,780 m.a.s.l. For *B. palmeri* the ecological niche model extended in the southwestern Mexican Plateau and few areas of the Sierra Madre del Sur and with altitudinal range between 1,175 to 2,050 m.a.s.l. In contrast, the niche of *B. cuneata* was smaller and projected in the central portion of the Trans-Mexican Volcanic Belt and some areas of the Sierra Madre del Sur and with higher altitudinal range between 1,750 to 2,400 m.a.s.l. (S2 Fig). The predicted niche overlap zone among the three species showed a sympatric area in the Bajío region, largely between the states of Michoacán and Guanajuato, with few small and fragmented areas across the Sierra Madre del Sur. The total extent of the overlap zone was 74,023 km$^2$, and accurately predicted the presence of the putative hybrids (Fig 4). The PCA-env was generated with the most important and common variables. The first axis explained 54% of the environmental variation (BIO1, BIO4 and BIO13) and the second axis explained the 25.5% of variation (BIO15) (S3 Fig)

The similarity test showed that the *B. palmeri* niche significantly predicted the niche of *B. cuneata* (D = 0.45, I = 0.67, $P < 0.05$), while the other species pairwise comparisons were not significant. Moreover, the *B. palmeri* niche significantly predicted the niche of the putative hybrids (D = 0.13, I = 0.36 $P < 0.05$), but *B. cuneata* predicted the niche of the putative hybrids with higher overlap values (D = 0.31, I = 0.54, $P < 0.05$). Neither *B. bipinnata* nor the putative hybrids could predict the niche of each other (D = 0.04, I = 0.20, $P > 0.05$) (Table 3, S3 Fig). The equivalency test indicated that the niches of *B. bipinnata* and *B. cuneata* (D = 0.26, I = 0.45, $P < 0.05$), and *B. cuneata* and *B. palmeri* were not significantly equivalent (i.e., divergent) (D = 0.45, I = 0.67, $P < 0.05$), while the niches of *B. bipinnata* and *B. palmeri* were equivalent (D = 0.46, I = 0.62, $P > 0.05$). The comparisons with respect to the putative hybrids and the parental species showed that the putative hybrids and *B. bipinnata* had not equivalent niches (D = 0.04, I = 0.20, $P < 0.05$), but the putative hybrids had equivalent niches with *B. palmeri* (D = 0.13, I = 0.36, $P > 0.05$) and *B. cuneata* (D = 0.31, I = 0.54, $P > 0.05$).

## Temporal projections of the hybrid zone

The ecological niche model during the warm period of the LIG (~ 130 kya) did not predict a niche overlap zone among the three *Bursera* species, but during the cooling conditions of the LGM (~ 21 kya) the CCSM4 and MIROC projections showed a niche overlap towards some disconnected areas in the south coast of the Pacific and the Sierra Madre del Sur. The MIROC model predicted a larger area of niche overlap (102,471 km$^2$) relative to the CCSM4 model (87,591 km$^2$). These two overlap niche areas did not coincide with presence of the putative hybrid records at the present. With the increase of temperature during the LH (~ 2.2 kya) under the MIROC model, the overlap niche zone shifted north towards the Bajío region and Sierra Madre del Sur, whereas for the CCSM4 model it was limited to the east of the Sierra Madre del Sur. For both models, the niche overlap zone decreased in extension relative to the LGM. For the LH, the putative hybrid records completely match the overlap niche zone predicted under the MIROC, and partially match under the CCSM4 (Fig 4).

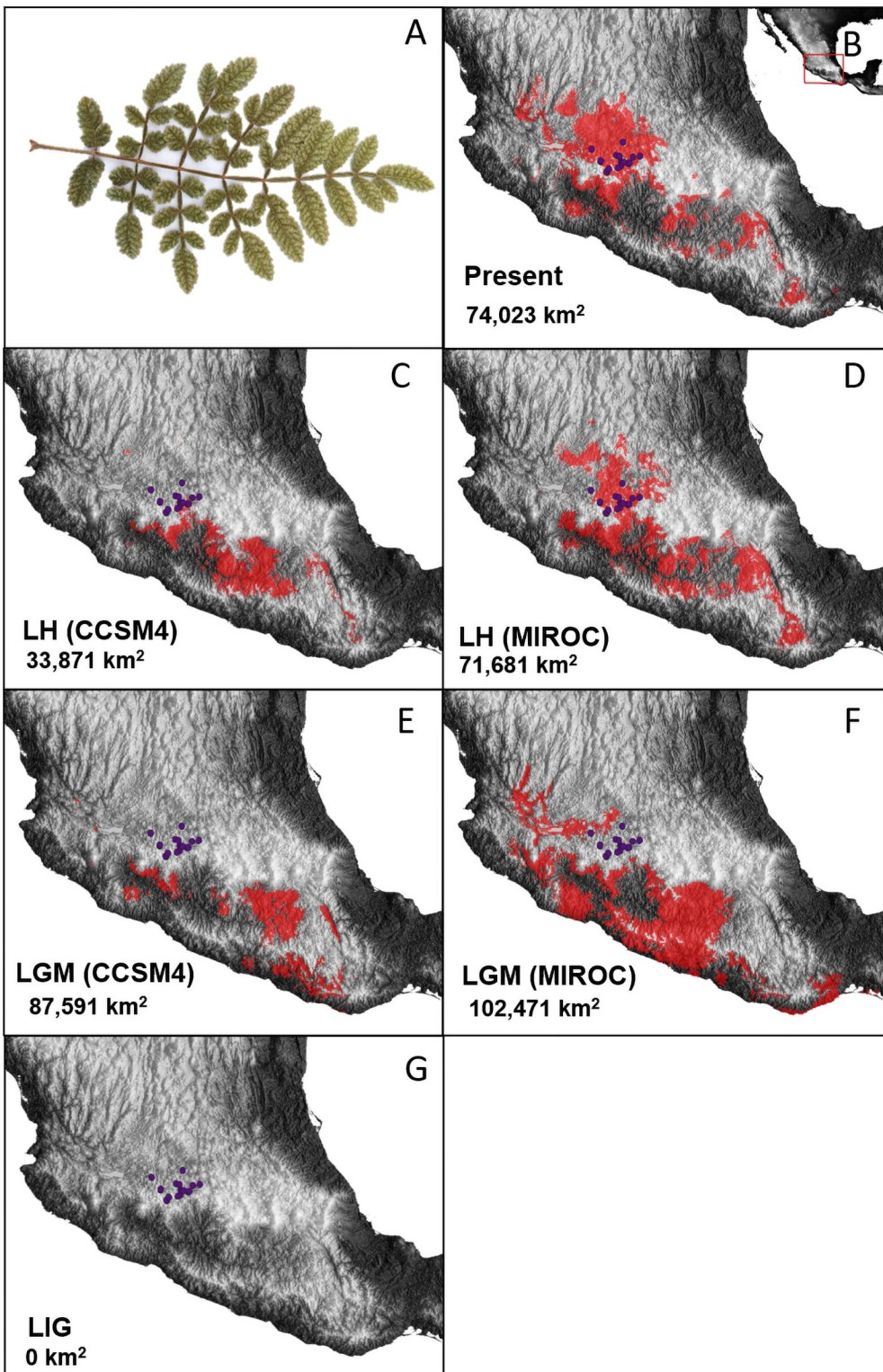

**Fig 4. Ecological niche projections of the overlap zone among *Bursera bipinnata*, *B. cuneata*, and *B. palmeri* at the present and past scenarios.** (A) Hybrid leaf sample, (B) present time, (C) late Holocene (LH, CCSM4), (D) late Holocene (LH MIROC), (E) Last Glacial Maximum (LGM, CCSM), (F) Last Glacial Maximum (LGM, MIROC) and (G) Last Interglacial (LIG). Red areas on the map indicate the predicted overlap niche zones, purple dots represent the occurrence of hybrid populations, numbers in km represent the total extension of the hybrid zone.

**Table 3. Pairwise niche overlap values of Schoener's D and Hellinger's I between the three *Bursera* species and their putative hybrids.**

| Group comparison | | Overlap metrics | | Niche similarity | | Niche equivalency |
|---|---|---|---|---|---|---|
| **a** | **b** | **Schoner's D** | **Hellinger's I** | **a → b** | **b → a** | |
| *B. bipinnata* | *B. cuneata* | 0.26 | 0.45 | NS | NS | NE |
| *B. bipinnata* | *B. palmeri* | 0.46 | 0.62 | NS | NS | Equivalent |
| *B. cuneata* | *B. palmeri* | 0.45 | 0.67 | Similar | NS | NE |
| Putative hybrids | *B. bipinnata* | 0.04 | 0.20 | NS | NS | NE |
| | *B. cuneata* | 0.31 | 0.54 | Similar | NS | Equivalent |
| | *B. palmeri* | 0.13 | 0.36 | Similar | NS | Equivalent |

a → b (niche a predicted by niche b), NS not similar ($P > 0.05$), NE not equivalent ($P < 0.05$).

For the future scenarios, the overlap niche zone was located mainly in the Bajío region. For CCSM4 global models, the predictions of the overlap zones were larger than for the MIROC models. For 2050, the CCSM4 scenario showed a relatively smaller overlap niche zone under the optimistic scenario (93,371 km$^2$, RCP 4.5), and a larger area under the pessimistic scenario (104,668 km$^2$, RCP 8.5). For 2070, the pessimistic and optimistic CCSM4 scenarios predicted very similar locations and extensions of the niche overlap zones. All 2050 and 2070 scenarios coincided with the presence of the putative hybrid records. In contrast to the CCSM4, the MIROC models predicted a decrease of the overlap niche zone from 2050 to 2070, although the location coincided with the CCSM4 models (Fig 5). According to the altitudinal density plots, a range shift of the niche overlap zone through time was evident: in the past (since the LGM) the hybrid zone occurred in the lowlands, whereas at the present and the future, the hybrid zone is shifting to greater elevational ranges (S4 and S5 Figs).

## Discussion

In this study, analysis of the ETS and PEPC nuclear sequences confirmed the hybrid origin of morphologically intermediate individuals in five populations with the co-occurrence of *B. bipinnata*, *B. palmeri* and *B. cuneata* in the Bajío region in Mexico. Results showed that the putative hybrids shared genetic ancestry from the three *Bursera* species, but with closer genetic affinity to *B. cuneata* and *B. bipinnata*. Ecological niche modeling accurately predicted the occurrence of known putative hybrid records and showed that the overlap zone among the three *Bursera* species, where potential hybridization may occur, extends in a larger area than previously thought. The putative hybrids did not show ecological divergence from *B. palmeri* and *B. cuneata* but did from *B. bipinnata*, while *B. palmeri* significantly predicted the niche of *B. cuneata*. Past and future ecological niche projections showed that the overlap zone likely occurred since the LGM (~ 21 kya) and have ascended altitudinally through time.

### Genetic evidence of interspecific hybridization in *Bursera*

Presence of morphologically intermediate individuals between *Bursera* species has been reported extensively from Baja California [20], the Bajío region [16, 19], the western Balsas [22], and to the Tehuacán-Cuicatlán region in Mexico [16, 73]. In the Bajío, Rzedowski and Guevara-Féfer [19] identified putative hybrids derived from *B. bipinnata* with *B. cuneata* and *B. palmeri*. Our findings from phylogenetic reconstructions revealed that these putative hybrids were intermixed within subclades of the three *Bursera* species. A similar pattern was observed by Weeks and Simpson [74] among hybridizing Hispaniolan *Bursera*. Moreover, the shared origin of the putative hybrids from the three *Bursera* species was clearer from the ETS

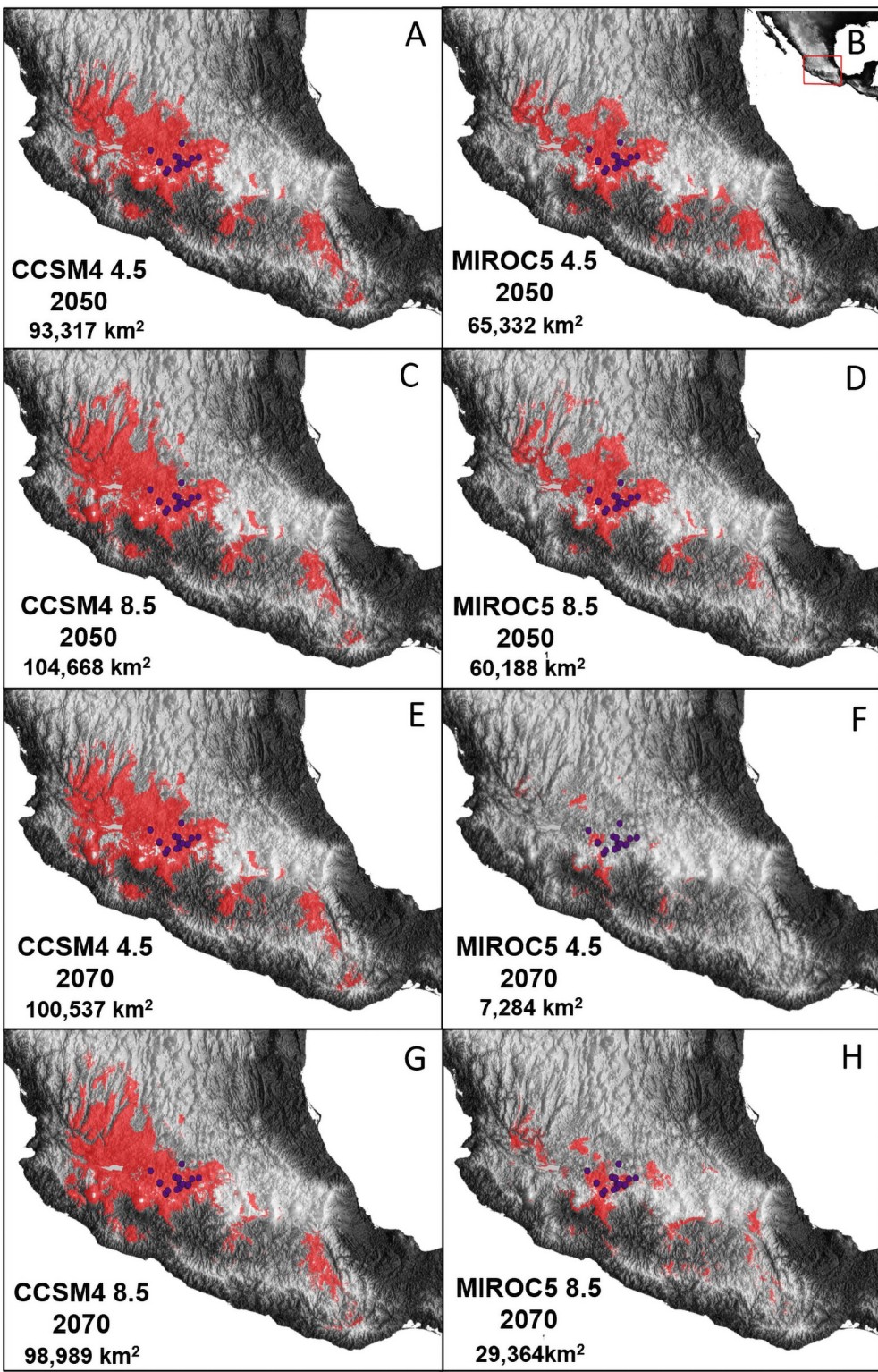

**Fig 5. Ecological niche predictions of the overlap niche zone among *Bursera bipinnata*, *B. cuneata*, and *B. palmeri* for future climatic scenarios: (A) 2050 optimistic scenario (RCP 4.5, CCSM4), (B) 2050 optimistic scenario (RCP 4.5, MIROC), (C) 2050 pessimistic scenario (RCP 8.5, CCSM4), (D) 2050 pessimistic scenario (RCP 8.5, MIROC), (E) 2070 optimistic scenario (RCP 4.5, MIROC), (F) 2070 pessimistic scenario (RCP 8.5, CCSM4), (G) 2070 pessimistic scenario (RCP 8.5, CCSM4), (H) 2070 pessimistic scenario (RCP 8.5, MIROC).** Red areas on the map indicate the predicted overlap niche zones, purple dots represent the occurrence of hybrid populations, and numbers in km represent the total extension of the hybrid zone.

and PEPC MJ haplotype networks, which showed that most common species haplotypes were shared in large proportion with the putative hybrids.

Interspecific hybridization is expected to be common in closely related species that have not developed strong reproductive isolation. For the *Bullockia* section, two natural groups were recognized based on the fruit and flower characteristics [22] and further confirmed with molecular data [74, 75]. Taxonomic identity of the three *Bursera* species here has been confirmed with morphological and molecular data [14, 16, 19]. According to the pairwise $F_{ST}$ estimates, genetic differentiation between *B. bipinnata* and *B. cuneata* ($F_{ST}$ = 0.47) is lower than between *B. bipinnata* and *B. palmeri* ($F_{ST}$ = 0.52). Likewise, the putative hybrids were more differentiated to *B. palmeri* relative to the other two species. This pattern was also evident from the DAPC plot, in which the putative hybrids were largely overlapped with *B. cuneata* and *B. bipinnata*. Our field observations and genetic data suggest that most of the putative hybrids were likely the result of interbreeding between *B. bipinnata* and *B. cuneata*, and to a lesser extent to *B. bipinnata* and *B. palmeri*, and then to *B. cuneata* and *B. palmeri*. Of the three species, *B. bipinnata* is the species that is known to hybridize with a larger number of *Bursera* species across its range [16].

Nothing is known about the development of reproductive barriers in *Bursera* which could help explain the propensity of some species to hybridize more frequently than others. Hybrid infertility is expected to limit the occurrence of backcrosses acting as a postzygotic reproductive barrier [76]. A recent study showed that pollen of these putative hybrids is fertile [77], which may facilitate the occurrence of backcrosses and $F_2$ hybrids. Results from BAPS revealed genetic admixture between putative hybrids and the three *Bursera* species, but it was noticeably the higher genetic affinity of some *B. bipinnata* and *B. cuneata* individuals to the putative hybrids. This result implies that patterns of hybridization among these *Bursera* species are more complex than previously thought. This observed genetic pattern may reflect backcrossing events or F2 hybrid generations in which morphologically intermediacy is no longer recognized. Although the ETS and PEPC sequences have proven to be useful in phylogenetic and hybridization studies in *Bursera* including the present study [21, 25], with these molecular markers is not feasible to unambiguously characterize $F_2$ hybrids and backcrosses scenarios. Moreover, we cannot completely rule out the possibility that shared haplotypes between parental species and putative hybrids were the result of incomplete lineage sorting or ancestral character retention [78].

Our estimates of genetic diversity showed that the putative hybrids had the highest levels of genetic diversity, including heterozygosity, expected from their mixed origins. Field observations have noted that these putative hybrids can form vigorous trees with abundant fruit and flower production [19], but there is no specific data on their local fitness. Hybridization is an important evolutionary force that can increase genetic diversity of the populations involved [12, 79]. If backcrosses are possible, hybridization can influence levels of genetic diversity and structure in parental species with consequences for local adaptation [80]. Further studies with a much higher number of unlinked molecular markers, i.e., single nucleotide polymorphism SNPs, in a large number of populations across the potential hybrid zone are needed to better characterize complex hybridization scenarios, including $F_1$, $F_2$ hybrids and of introgression events, and to understand the role of interspecific hybridization in population genetic composition and adaptive introgression.

## The overlap niche zone and ecological divergence

Our results from ecological niche modeling showed a wide overlap niche zone among the three *Bursera* species that extends across the Bajío region comprising mostly the states of

Jalisco, Michoacán, Guanajuato, and Querétaro. This ecological overlap zone accurately predicted the occurrence of known putative hybrid records, which provides evidence to suggest that it corresponds to a potential hybrid zone [10]. We found that the most important environmental variables associated with the potential hybrid zone were temperature and precipitation, the ecological factors that have been associated to the distribution of *Bursera* species in the TDF [53, 81]. This potential hybrid zone largely coincides with the habitat of *B. palmeri* and *B. cuneata*. It is important to emphasize that hybridization rates and hybridization success likely vary across species ranges where they co-occur [82, 83]. Our model predictions on the extension of the overlap niche zone should be interpreted cautiously and not as strong evidence of hybrid presence as few hybrid records exist. Hence, it would be key to verify and corroborate the presence of hybrids at other sites across the potential hybrid zone where *Bursera* hybrids have not been reported, but which are predicted to exist.

The similarity test showed that *B. cuneata* had an ecological niche similar to *B. palmeri*, but the equivalence test showed that these species had divergent niches, while *B. bipinnata* only showed ecological divergence with *B. cuneata*. Among all *Bursera* species, *B. cuneata* occurs at higher altitudinal ranges (1850 to 2500 m.a.s.l.), usually associated with transition zones between the TDF and pine-oak forests [19]. These results suggest that ecological divergence might be a mechanism promoting reproductive isolation between *B. cuneata* and *B. bipinnata* which are likely the species that hybridize more frequently, and the most genetically similar. Ecological divergence has been suggested to contribute to the establishment of reproductive barriers between hybridizing species [84, 85] for several taxa including insects [27], reptiles [86, 87], mammals [88], and invasive plants [89].

Ecological divergence of the hybrids with respect to the parental species can be a signature of their ecological adaptation across hybrid zones [5] and their potential to facilitate the development of novel adaptations to changing environments [90, 91]. We found that putative hybrids showed ecological divergence to *B. bipinnata*, but not to *B. cuneata* and *B. palmeri*, despite that *B. palmeri* showed the highest genetic differentiation to the putative hybrids. The putative hybrids exhibited the highest niche similarity with *B. cuneata*. The absence of ecological divergence of the putative hybrids with *B. cuneata* further confirms their high relationship and suggests that hybrids might not form and establish independent populations from this parental species (which we have observed in the field). However, signatures of ecological divergence might not be detected at this regional scale. Both endogenous and exogenous sources of selection on hybrids can vary substantially across a hybrid zone which can differentially influence hybridization and introgression rates and hybrid fitness at large and local scales [92–94]. For plants, it has been observed that hybridization rates can be influenced by pollinator behavior [95], reproductive phenology [96], ecological disturbance [97], and elevational ranges [92]. The patchy distribution of the TDF and the high landscape heterogeneity in which these *Bursera* species currently persist [34], may impose varying selective forces at short spatial scales with implications in local fitness. Future research is needed to explore the role of the ecological context on hybridization rates and hybrid fitness by increasing the number of populations analyzed across the potential hybrid zone.

## Temporal dynamics of the hybrid zone

The temporal overlap niche projections using past bioclimatic variables predicted the occurrence of a potential hybrid zone during the Last Glacial Maximum (~ 21 kya), although its distribution was largely located in southern Mexico in the Sierra Madre del Sur. Later during the warm conditions of the late Holocene (~ 2.3 kya), this potential hybrid zone shifted northward, but it contracted in extension. In this period, both models predicted the presence of the

current hybrid records. In Mexico, during the interglacial and glacial climatic cycles of the Pleistocene, forest species experienced repeated expansions and contractions in response to climatic changes [98]. There is scarce information on the historical temporal changes of TDF species during the Pleistocene in Mexico. Based on ecological niche modeling for areas of endemism in *Bursera*, Gámez et al. [54] found elevational changes of *Bursera* species from the LGM to the present in response to climate warming and that may be related to limited historical dispersal [14]. The potential hybrid zone location during the LGM for the MIROC model corresponds to an area known as the Western Balsas, while for the CCSM model, the potential hybrid zone was predicted in the Eastern Balsas. Both areas were suggested as refugia during the cooler and wetter conditions of the LGM [54]. From the late Holocene to the present there was an increase in the extension of the potential hybrid zone. Overall, future scenario projections under the CCSM model showed a potential increment of the suitability for the hybrid zone for 2050 and 2070 relative to the present. For both circulation models, the predictions indicated the increase in elevational ranges, but with a larger extent under the CCSM4 model relative to MIROC, which spatial extent decreases by 2070. The increase in elevational ranges coincided with the observed trends of the TDF to ascend altitudinally in response to climatic changes [54, 99] which likely will create the conditions for other *Bursera* species to come into contact. This may increase the opportunities for interspecific gene exchange as it has been predicted for several taxa [11, 100] and with important consequences for TDF biodiversity including species losses and gains at local scales [54].

## Conclusion

This study provides the first insight into existence of a potential hybrid zone among three Mexican *Bursera* species (*Bullockia* section) based on molecular evidence and ecological niche modeling. The examination of the temporal dynamics of the hybrid zone in response to climatic changes provided useful information in understanding the role of environmental factors in shaping the formation and maintenance of potential hybrid zones in this plant taxa with a high number of co-occurring endemic species.

## Supporting information

**S1 Fig. BEAST phylogenetic tree of the concatenated ETS and PEPC nuclear genes.** The tree shows the relationships between the tree *Bursera* species and the putative hybrids. Outgroup species were *B. simaruba*, *B. kerberi*, and *B. fagaroides*. Values above nodes represent posterior probabilities.
(PDF)

**S2 Fig. Predicted suitable areas for each *Bursera* species through ecological niche modeling.** The model was constructed based on the best calibration based on Kuenm. Green dots represent the training set and blue dots the testing set. The black and gray shading represent an altitudinal gradient with lighter areas being altitudinally higher.
(PDF)

**S3 Fig. Results from the similarity test between the putative hybrid and each of the parental *Bursera* species.** The left panel shows the environmental space of *Bursera palmeri* (orange), *B. cuneata* (blue) and *B. bipinnata* (green) and the putative hybrids (purple). The gray area denotes the overlap area. The right panel shows the distribution of observed vs expected values of Schoener's D, the blue lines denote the range of 95% of expected values and the red lines the observed values.
(PDF)

**S4 Fig. Density plots for the CCSM4 models showing the altitudinal range values predicted for the overlap niche zone (i.e., hybrid zone) for each climatic niche scenario.** Scenarios: LGM Last Glacial Maximu, LH Late Holocene. Values of the x-axis denotes the elevational ranges.
(PDF)

**S5 Fig. Density plots for the MIROC models showing the altitudinal range values predicted for the overlap niche zone (i.e., hybrid zone) for each climatic niche scenario.** Scenarios: LGM Last Glacial Maximum, LH Late Holocene. Values of the x-axis denotes the elevational ranges.
(PDF)

**S1 Table. Species occurrence used for the niche modelling.**
(XLSX)

**S2 Table. Pairwise $F_{ST}$ (below) and $G_{ST}$ (above) genetic distances among the three *Bursera* species and the putative hybrids.** All comparisons were statistically significant $P$ = 0.0001.
(PDF)

## Acknowledgments

For assistance in the field and sample preparation we thank Benjamín Castillo Ponce, Bruno A. Gutiérrez Becerril, Stephanie Aguilera López, Tania Andrade Ortiz, and Víctor Reyes Pino, and to several field assistants from local communities. A sampling permit was granted by SEMARNAT SGPA/DGGFS/712/1062/18. We are thankful to two external reviewers and the editor for comments that helped us significantly improve the quality of our manuscript.

## Author Contributions

**Conceptualization:** Yessica Rico, Andrés Lira Noriega.

**Data curation:** Eduardo Quintero Melecio.

**Formal analysis:** Eduardo Quintero Melecio, Yessica Rico.

**Funding acquisition:** Yessica Rico.

**Investigation:** Eduardo Quintero Melecio, Yessica Rico.

**Project administration:** Yessica Rico.

**Resources:** Yessica Rico, Andrés Lira Noriega, Antonio González Rodríguez.

**Supervision:** Yessica Rico, Andrés Lira Noriega, Antonio González Rodríguez.

**Validation:** Yessica Rico.

**Writing – original draft:** Eduardo Quintero Melecio, Yessica Rico.

**Writing – review & editing:** Eduardo Quintero Melecio, Yessica Rico, Andrés Lira Noriega, Antonio González Rodríguez.

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
