## [Decision Letter · Decision Letter 0]

10 Aug 2021

PONE-D-21-19713

Molecular evidence and ecological niche modeling reveal an extensive hybrid zone among three Bursera species (Section Bullockia)

PLOS ONE

Dear Dr. Rico,

Thank you for submitting your manuscript to PLOS ONE. After careful consideration, we feel that it has merit but does not fully meet PLOS ONE’s publication criteria as it currently stands. Therefore, we invite you to submit a revised version of the manuscript that addresses the points raised during the review process.

Two external referees and me have now completed a review of this manuscript. I am sorry that it as taken this long to complete the first round of reviews, but reviewers were hard to find, and extra time was needed. I think both reviewers have provided copious constructive criticisms that will help the authors improve their analysis and interpretation of their results. One reviewer suggested that you make the R code available for your readers. This is necessary when unique scripts are produced as part of the methods in your study. Please carefully consider these comments and describe in detail how you integrate these new analyses in your study.

We look forward to receiving your revised manuscript.

Kind regards,

William J. Etges

Academic Editor

PLOS ONE

“This study was funded by Consejo Nacional de Ciencia y Tecnología (CONACYT) through the Ciencia Básica CB2016-283237 project to YR and a Master’s scholarship to EQM (MM18019).”

Funding information should not appear in the Acknowledgments section or other areas of your manuscript. We will only publish funding information present in the Funding Statement section of the online submission form.

 “This study was funded by Consejo Nacional de Ciencia y Tecnología (CONACYT www.conacyt.gob.mx) through the Ciencia Básica CB2016-283237 project to YR and a Master’s scholarship to EQM (MM18019).

The funders had no role in study design, data collection and analysis, decision to publish, or preparation of the manuscript”

3. We note that you have referenced (Rico et al. unpublished data) which has currently not yet been accepted for publication. Please remove this from your References and amend this to state in the body of your manuscript: (Rico et al. unpublished data) as detailed online in our guide for authors

http://journals.plos.org/plosone/s/submission-guidelines#loc-reference-style.

Reviewers' comments:

Reviewer's Responses to Questions

**Comments to the Author**

1. Is the manuscript technically sound, and do the data support the conclusions?

Reviewer #1: Partly

Reviewer #2: Yes

2. Has the statistical analysis been performed appropriately and rigorously? 

Reviewer #1: N/A

Reviewer #2: Yes

3. Have the authors made all data underlying the findings in their manuscript fully available?

Reviewer #1: Yes

Reviewer #2: Yes

4. Is the manuscript presented in an intelligible fashion and written in standard English?

Reviewer #1: Yes

Reviewer #2: Yes

5. Review Comments to the Author

Reviewer #1: The authors sampled some putative hybrids and individuals from three parental species of *Bursera* (*B. bipinnata, B. palmeri* and *B. cuneata*). The aims of the study were to identify whether individuals with intermediate morphological characteristics are hybrids from three *Bursera* species; to identify the closest genetic affinity of these putative hybrids; to characterize and compare the ecological niche of hybrids and the three parental species; and to project the geographic dynamics of the hybrid zone through time. Sampling was conducted in nine sites, four of which are suspected to be hybrid zones. The authors amplified five chloroplast DNA regions and three nuclear regions. The chloroplast sequences and one nuclear marker for a subset of individuals did not show polymorphism and were not used for further analyses. All analyses were conducted for ETS and PEPC nuclear regions, which were sequenced for 101 individuals. Also, the authors produced species distribution models and past projections to the Last Interglacial, Last Glacial Maximum and Late Holocene and to the future 2050 and 2070 using two RCP (4.5 and 8.5) and two AOGCM (MIROC and CCSM). Haplotypes were shared between putative hybrids and parental species, but also among parental species. Putative hybrids were genetically more similar to *B. palmeri*. Ecological niche models adequately predicted the hybrid zone. Past projections suggest that the hybrid zone formed around the LGM and has persisted since. Future projections suggest for one AOGCM suggest expansion of the hybrid zone and for the other a contraction, particularly for 2070.

The study is interesting, and data is adequate for this type of analysis. Analyses of species distribution models are adequate and robust. Nevertheless, for sequences of nuclear regions it is recommended to clone sequences or at least use a program such as PHASE (Garrick et al. 2010 BMC Evolutionary Biology 10:118) to resolve haplotype phases for a robust haplotype designation, and also to correctly identify heterozygote individuals. This is particularly important because every result will be affected by the correct characterization of haplotypes. In addition, it is important to conduct recombination analyses and eliminate recombinant sequences in phylogenetic reconstructions to avoid biases and artifacts in the tree topology (Holmes et al. 1999 Molecular Biology and Evolution 16(3):405-409; Hare 2001 Trends in Ecology and Evolution 16(12):700-706; Posada and Crandall 2002 Journal of Molecular Evolution 54:396-402). Moreover, a recombination analysis could support the designation of putative hybrids as true hybrids because we could expect that these individuals will show mixed haplotypes.

According to the methods, they authors focused their analyses on nuclear haplotypes, which is adequate for genealogical reconstruction through BI and MJ networks. In this sense, the results from the PCA do support that putative hybrids could be true hybrids. To help the reader interpret the results from the genealogy and MJ network it would be useful if in the introduction the authors explain which would be the expected topology for the genealogy and MJ network that would support the hybrid origin of putative hybrid individuals. Later in the manuscript (discussion section), the authors should back their arguments supporting the hybrid origin of putative hybrids in light of those expected results. Nuclear sequence data could be used to estimate heterozygosity, gene flow or admixture (see for example Moreno-Letelier et al. 2014 New Phytologist 203:335-347). Thus, providing additional support to the hybrid origin of putative hybrids.

Other comments

Abstract. Please provide some detail regarding the methods, i.e. type of data used for molecular analyses (nuclear data?) and species distribution models past and future projections (time periods?).

Introduction. Consider modifying the organization of the information. Some ideas feel disconnected. At the end of the introduction, please provide an objective that clearly justifies the use of future projections for ecological niche models.

Line 1. Please change “~100 species of shrub and trees of the tropical dry forests” to “~100 shrub and tree species of the tropical dry forests”.

Line 9. Please specify which type of sequences and from which genomic region or regions were used for molecular analyses.

Lines 35-37. Please revise this sentence. Hybridization in animals is more frequent than previously thought; therefore, I recommend changing this sentence to “Hybridization is common in plants and there are still many groups of plant taxa in which hybridization is poorly understood”.

Line 38. Please change “genera”, which is plural, to “genus”, the singular form of the term.

Line 77. Revise “The TDF of the Bajio at the central west of Mexico”, consider “The TDF of the Bajio in Central-West Mexico”.

Lines 77-89. Sentence about TDF distribution and reports of Bursera hybrids in the Bajio feel disconnected. Please review and consider moving up, to line 66 “Specifically, in the TDF of the Bajio… , while no information exists on the occurrence of hybrid zones for any Mexican Bursera.”

Also, consider moving up “The TDF of the Bajio in Central-West Mexico, was one of the most widespread ecosystem in the region, which currently is disappearing by anthropogenic land use changes, such as increase in livestock grazing, agriculture, and urbanization (37)” to line 49 where you talk about the TDF.

Line 86. Please change “expected” to “considered”.

Lines 86-89. Please review this sentence. It is too long and it is difficult to follow the main idea.

Lines 95-97. Consider adding “and its maintenance?” at the end of the last question “Can we spatially and environmentally predict the formation of the hybrid zone, and which is the role of past and future climatic conditions on its formation”. Past climate change may have had a role in the formation of the hybrid zone, but future climate change will not have this role because the hybrid zone already exists; nevertheless, future climate change might play a role in the continuity of such hybrid zone.

Line 100. Target species and sampling.

Line 101-124. Please mention the ploidy reported for Bursera species. Are they diploid or polyploid?

Line 106. Change “Leave” for “Leaf”.

Lines 143-150. I strongly suggest depositing cpDNA sequences in NCBI even if the authors did not find genetic variation. This could be useful information for other analyses such as phylogenetic studies.

Lines 153-168. Please describe how you treated heterozygous individuals in your analyses. Were BI and haplotype network analyses conducted on haplotypes or genotypes? How did you identified or coded putative hybrids?

Lines 170-171. Please provide a justification or the rationale for obtaining three MJ networks, one for each analyzed gene and one with concatenated data.

Line 304. Please change “high-supported” to “well-supported”.

Line 311. Unfortunately, I could not access the supporting information. I recommend moving the BI tree to main text; particularly, if any inference regarding the identity of putative hybrids is being made from this result.

Line 332. Authors mention that “Estimates of genetic diversity for cpDNA concatenated matrix…”; nevertheless, in the methods section (lines 142-144) they mention that “The five cpDNA regions an dNIA-i3 resulted non-polymorphic in a subsert of DNA samples for each species. Thus, our results are based in the polymorphism of the ETS and PEPC genes”. Please check this information.

Line 434. Please change “…the overlap niche zone” to “…the overlap zone”.

Line 436. Please change “…did to…” to “…did from…”.

Lines 435-436. Please also mention that “B. palmeri significantly predicted the niche of B. cuneata”. Also, it is interesting to note, and worth highlighting, that putative hybrids showed niche similarity to B. palmeri but these taxa showed the highest genetic differentiation.

Line 459. Change “reportedly” to “reported”.

Line 464-465. In “Our findings from the phylogenetic reconstruction based on nuclear markers revealed that these putative hybrids were intermixed between subclades of the three Bursera species”. Please, provide a reference supporting this statement, and explain what the expected topology would be when a hybrid zone occurs (Soltis et al. 2008. Systematic Botany 33(1):7-20; Okuyama et al., 2005. Molecular Biology and Evolution 22(2):285-296). Same thing for lines 469-470 regarding haplotype networks; why is it expected that the haplotypes from putative hybrids will be intermediate links between parental species? What would be the difference between putative hybridization and the clinal variation mentioned in lines 456-457?

Lines 468. It is worth considering in the discussion that, given the lack of polymorphism in five regions of cpDNA, we cannot rule out that shared nuclear haplotypes between the three species and putative hybrids might be related to ancestral character retention or incomplete lineage sorting. In this sense, I recommend conducting recombination tests and also I recommend codifying genotypes for each individual which will allow conducting estimating allele frequencies, observed heterozygosity and test for admixture.

Line 510. Change “area” to “are”.

Line 582-589. Regarding the results for future projections of the species distribution models, what could be the potential implications of the predicted expansion or contraction of the hybrid zone -depending on the AOGCM- for the analyzed taxa?

Reviewer #2: Title: Molecular evidence and ecological niche modeling reveal an extensive hybrid zone among three *Bursera* species (Section Bullockia)

Manuscript ID: PONE-D-21-19713

Reviewer Summary: Quintero-Melecio et al. use a combination of niche modeling and population genetic analysis to assess and describe the presence of a hybrid zone among three species of Bursera in the section Bullockia. Their overall findings support the existence of an ecologically divergent set of hybrids with *B. cuneata* being the closest parental species to these hybrids. Niche modeling and patterns of niche divergence were further investigated to quantify the presence of the hybrid zone, including historical and future projections. These analyses supported a model of ecologically divergent hybrids, with an extensive current distribution.

Overall Impressions: Overall, I find the results in the manuscript to be compelling. The presentation is clear and the methodological choices are sensible, although they could use further justifications, assessment of assumptions, and descriptions in places (see below). WIth that said, I think some of the main conclusions could be bolstered in places. For example, is there no other possible putative hybrid occurrence point or points (from an herbarium or observational) to validate the extensive hybrid zone inferred from parental species overlap (as in Figure 4B)? Further comments are given below.

Comments:

Pg. 4, lines 53-55. While hybrids are often phenotypically “intermediate” to “parent” species, they do not necessarily have to be so, which is consistent with some of your results from BAPS. I think there was a missed opportunity later in the manuscript to highlight the importance of genetic data, even from only two nuclear loci, to working out complex hybridization scenarios.

Pgs 7 - 8, lines 127 - 150. How were heterozygous sites dealt with during sequence analysis (i.e., your sampled tree is a heterozygote for a polymorphic site)? For example, other studies have cloned PCR products for ETS in other *Bursera* species (see Weeks & Simpson 2004 Am. J. Bot. 91: 976-984). If they were ignored then would this not then create a bias? If these sites were called polymorphic then what was the procedure by which they were deemed true positives (e.g., comparing peak heights, areas, etc. on the chromatograms)?

Pg. 10 line 197. What was the pairwise distance used in AMOVA? Was it simple p-distances or some transformation of those assuming a finite sites model? If p-distances, why not use pairwise distance from the best-fit model of sequence evolution from JMODELTEST?

Pg. 10, lines 193-200. Are the AMOVAs and Fst/Gst analyses really needed? My concern is that these methods, especially with complex hierarchies, are really designed for nested categories (i.e., trees nested in populations and populations nested in species), but with hybridization do you not have a different kind of categorical organization (something like a partially crossed design that violates the assumptions of the method where trees that are hybrids are really more like an interaction between two species rather than an independent category of the same meaning as the “parent” species categories)?

Pg. 11, lines 212-214. How were the thresholds of 20 km and 15 km arrived at for spatial thinning? I like that the threshold was lower for species with the smaller distributional area, but were there other criteria for choosing these values? Were the hybrids also thinned and, if so, why were they thinned?

Pgs. 13 - 14, lines 273-275. Unequal sampling effort is definitely a problem given the variation in sample sizes for the “parent” species and the hybrids, but this is not the same as small sample sizes for the putative hybrids (n = 19). I think it would be helpful for the general reader to have a few arguments about why such small and spatially localized samples (see Fig. 4) are sufficient for this analysis.

Pg. 14, lines 285-286. By equating the hybrid calibration area to the area of niche overlap of the “parent” species, are you not assuming that hybridization is successful anywhere and with any individuals from each species in this overlap area? Is this reasonable given that there are examples of variable hybridization success across ranges of species (e.g., Mandeville et al. 2017 Evol. Letters 1: 255-268) and that populations of trees often have strong patterns of local adaptation? Maybe some description and justification of this assumption is needed? This also applies to the measurement of the hybrid zone as given in Figure 4 and elsewhere because overlap of the “parent” species is being used as a measurement of hybrid presence.

Pg. 13 lines 254 - 267. Was there a particular reason that ensemble models were not presented, maybe an ensemble across CCSM4.0 and MIROC5 for each RCP? This might help readers with the overall results from the ENMs.

Pg. 14, lines 296-297. For the niche equivalency and similarity tests using ecospat, did you specify the alternative flag in each function (i.e. -alternative = “greater” is niche conservatism and -alternative = “lower” is niche divergence)? If so, which direction did you specify and why? Lastly, why not use the niche overlap test?

Pg. 24, lines 512-516. It is not that you need markers with higher levels of polymorphism. It is that you would need many more unlinked markers, which is what SNPs derived from one of many different forms of GBS would do for you. Even then it may be difficult to parse hybrids into F1, F2, BC, etc. I think this sentence needs a slight tweak to emphasize the number of unlinked markers rather than “higher polymorphism”. Lastly, the point about sampling more populations of hybrids is valid, but is it not also sampling more of the geographical area of potential hybridization?

Editorial Suggestions:

Pg. 24, line 510. I believe “area” should be “are”.

It is my policy to sign all of my reviews: Andrew J. Eckert (aeckert2@vcu.edu)

6. PLOS authors have the option to publish the peer review history of their article (what does this mean?). If published, this will include your full peer review and any attached files.

Reviewer #1: No

Reviewer #2: **Yes: **Andrew J. Eckert

---

## [Author Response · Author response to Decision Letter 0]

27 Sep 2021

Reviewers' comments:

Reviewer #1: 

The authors sampled some putative hybrids and individuals from three parental species of Bursera (B. bipinnata, B. palmeri and B. cuneata). The aims of the study were to identify whether individuals with intermediate morphological characteristics are hybrids from three Bursera species; to identify the closest genetic affinity of these putative hybrids; to characterize and compare the ecological niche of hybrids and the three parental species; and to project the geographic dynamics of the hybrid zone through time. Sampling was conducted in nine sites, four of which are suspected to be hybrid zones. The authors amplified five chloroplast DNA regions and three nuclear regions. The chloroplast sequences and one nuclear marker for a subset of individuals did not show polymorphism and were not used for further analyses. All analyses were conducted for ETS and PEPC nuclear regions, which were sequenced for 101 individuals. Also, the authors produced species distribution models and past projections to the Last Interglacial, Last Glacial Maximum and Late Holocene and to the future 2050 and 2070 using two RCP (4.5 and 8.5) and two AOGCM (MIROC and CCSM). Haplotypes were shared between putative hybrids and parental species, but also among parental species. Putative hybrids were genetically more similar to B. palmeri. Ecological niche models adequately predicted the hybrid zone. Past projections suggest that the hybrid zone formed around the LGM and has persisted since. Future projections suggest for one AOGCM suggest expansion of the hybrid zone and for the other a contraction, particularly for 2070.

The study is interesting, and data is adequate for this type of analysis. Analyses of species distribution models are adequate and robust. Nevertheless, for sequences of nuclear regions it is recommended to clone sequences or at least use a program such as PHASE (Garrick et al. 2010 BMC Evolutionary Biology 10:118) to resolve haplotype phases for a robust haplotype designation, and also to correctly identify heterozygote individuals. This is particularly important because every result will be affected by the correct characterization of haplotypes. In addition, it is important to conduct recombination analyses and eliminate recombinant sequences in phylogenetic reconstructions to avoid biases and artifacts in the tree topology (Holmes et al. 1999 Molecular Biology and Evolution 16(3):405-409; Hare 2001 Trends in Ecology and Evolution 16(12):700-706; Posada and Crandall 2002 Journal of Molecular Evolution 54:396-402). Moreover, a recombination analysis could support the designation of putative hybrids as true hybrids because we could expect that these individuals will show mixed haplotypes.

R= We are very thankful for your comments that were key to improve the quality of our work. We missed these aspects in the first submission, but now we have corrected this problem. As suggested, we coded unambiguous sites for each region (as visualized carefully in the chromatograms) using the IUPAC nomenclature and then we used the PHASE software to obtain the gametic phase of heterozygous individuals (phased sequences) (See lines 164-167). With the phased sequences we conducted all genetic analysis (including an analysis of admixture and expected heterozygosity), and as expected a large number of putative hybrids were heterozygotes sharing haplotypes of parental species. The genbank accession numbers for sequences have been updated with the IUPAC code.

Moreover, we conducted a recombination analysis using the PHI-test and RDP (7 alternative methods) (See lines 168-173), which showed evidence of recombination, but only three sequences had significant evidence of recombination (See 333-338). For the phylogenetic reconstructions, the goal was to observe with which parental species the putative hybrids are more related and to make evident their mixed origin, and not to obtain the species relationships per se, in this case, the recombination events are informative of the process under study. For this, in addition to perform the BI analysis, we conducted a phylogenetic network using SplitsTree to obtain a reticulated network, which is more appropriate for the context of our study. The resulting network showed a high reticulated network and the higher intermixing of putative hybrids among major clades (See lines 344-348). 

According to the methods, they authors focused their analyses on nuclear haplotypes, which is adequate for genealogical reconstruction through BI and MJ networks. In this sense, the results from the PCA do support that putative hybrids could be true hybrids. To help the reader interpret the results from the genealogy and MJ network it would be useful if in the introduction the authors explain which would be the expected topology for the genealogy and MJ network that would support the hybrid origin of putative hybrid individuals. Later in the manuscript (discussion section), the authors should back their arguments supporting the hybrid origin of putative hybrids in light of those expected results. Nuclear sequence data could be used to estimate heterozygosity, gene flow or admixture (see for example Moreno-Letelier et al. 2014 New Phytologist 203:335-347). Thus, providing additional support to the hybrid origin of putative hybrids.

R= Thanks for your suggestions, as our analyses and interpretations are based on more than the BI and MJ networks, we decided not to mention the expectation of these BI and MJ in the introduction, because we will have to do so for all analyses. However, in the method sections we mentioned the expectation for the topology of the MJ network (See lines 189-197) and in the discussion we make clearer how the evidence support the origin of putative hybrids (See lines 531-538).

Moreover, as suggested, we conducted an admixture analysis using BAPS, which is more appropriate for sequence data and also calculated heterozygosity. Putative hybrids showed the higher admixture and heterozygosity, which further confirm their mixed origin (Lines 382-389). 

Other comments

Abstract. Please provide some detail regarding the methods, i.e. type of data used for molecular analyses (nuclear data?) and species distribution models past and future projections (time periods?).

R= done (line 9-10)

Introduction. Consider modifying the organization of the information. Some ideas feel disconnected. At the end of the introduction, please provide an objective that clearly justifies the use of future projections for ecological niche models.

R= We have modified the order of some ideas (as you suggested below). We also clarifiy the use of ecological niche modeling in objectives (lines 90-92)

Line 1. Please change “~100 species of shrub and trees of the tropical dry forests” to “~100 shrub and tree species of the tropical dry forests”.

R= done

Line 9. Please specify which type of sequences and from which genomic region or regions were used for molecular analyses.

R= done

Lines 35-37. Please revise this sentence. Hybridization in animals is more frequent than previously thought; therefore, I recommend changing this sentence to “Hybridization is common in plants and there are still many groups of plant taxa in which hybridization is poorly understood”.

R= done

Line 38. Please change “genera”, which is plural, to “genus”, the singular form of the term.

R= done

Line 77. Revise “The TDF of the Bajio at the central west of Mexico”, consider “The TDF of the Bajio in Central-West Mexico”.

R= done

Lines 77-89. Sentence about TDF distribution and reports of Bursera hybrids in the Bajio feel disconnected. Please review and consider moving up, to line 66 “Specifically, in the TDF of the Bajio… , while no information exists on the occurrence of hybrid zones for any Mexican Bursera.”Also, consider moving up “The TDF of the Bajio in Central-West Mexico, was one of the most widespread ecosystem in the region, which currently is disappearing by anthropogenic land use changes, such as increase in livestock grazing, agriculture, and urbanization (37)” to line 49 where you talk about the TDF.

R= The ideas relating to the TDF of Mexico and the importance of Bursera species were moved after the ideas relating to hybridization (See lines 71-77)

Line 86. Please change “expected” to “considered”.

R= done

Lines 86-89. Please review this sentence. It is too long and it is difficult to follow the main idea.

R= we have restructured the sentence (See lines 81-84).

Lines 95-97. Consider adding “and its maintenance?” at the end of the last question “Can we spatially and environmentally predict the formation of the hybrid zone, and which is the role of past and future climatic conditions on its formation”. Past climate change may have had a role in the formation of the hybrid zone, but future climate change will not have this role because the hybrid zone already exists; nevertheless, future climate change might play a role in the continuity of such hybrid zone.

R= agreed, we have added the suggestion to be more precise (see line 92)

Line 100. Target species and sampling.

R= done

Line 101-124. Please mention the ploidy reported for Bursera species. Are they diploid or polyploid?

R= There is no information on the chromosome number in these species, but they are expected to be diploid as in other Bursera (See lines 101-102). We have attempted to determine the chromosome number in these species, and preliminary we can confirm they are diploids. 

Line 106. Change “Leave” for “Leaf”.

R= done

Lines 143-150. I strongly suggest depositing cpDNA sequences in NCBI even if the authors did not find genetic variation. This could be useful information for other analyses such as phylogenetic studies.

R= agreed, will be available in GenBank. The sequence data has been already submitted and we are waiting for the ID assignation from the Genbank. 

Lines 153-168. Please describe how you treated heterozygous individuals in your analyses. Were BI and haplotype network analyses conducted on haplotypes or genotypes? How did you identified or coded putative hybrids?

R= Phylogenetic and network analysis were conducted with haplotypes. We coded unambiguous DNA sites (ej. A/G = R) using the IUPAC nomenclature and then used the PHASE software to phase sequences as suggested. (see lines 151; 164-167)

Lines 170-171. Please provide a justification or the rationale for obtaining three MJ networks, one for each analyzed gene and one with concatenated data.

R= We decided to provide only the MJ network for each marker separately, here the rationale is that we would expect that the most frequent haplotypes of the parental species would be shared with the putative hybrids irrespective of the nuclear marker. This was added in lines 195-197. The figure 2A-C is a composite of the neighbornet network and the MJ for each marker, which all together showed the high intermixing of putative hybrids with haplotypes of the parental species. 

Line 304. Please change “high-supported” to “well-supported”.

R= done

Line 311. Unfortunately, I could not access the supporting information. I recommend moving the BI tree to main text; particularly, if any inference regarding the identity of putative hybrids is being made from this result.

R= We decided to keep the BI as supplemental, but instead we implemented a neighbornet tree to better picture reticulate relationships. This is now included in figure 2

Line 332. Authors mention that “Estimates of genetic diversity for cpDNA concatenated matrix…”; nevertheless, in the methods section (lines 142-144) they mention that “The five cpDNA regions an dNIA-i3 resulted non-polymorphic in a subsert of DNA samples for each species. Thus, our results are based in the polymorphism of the ETS and PEPC genes”. Please check this information.

R= Sorry for the mistake, it should read nuclear concatenated matrix, we have corrected the sentence (line 331)

Line 434. Please change “…the overlap niche zone” to “…the overlap zone”.

R= done

Line 436. Please change “…did to…” to “…did from…”.

R= done

Lines 435-436. Please also mention that “B. palmeri significantly predicted the niche of B. cuneata”. Also, it is interesting to note, and worth highlighting, that putative hybrids showed niche similarity to B. palmeri but these taxa showed the highest genetic differentiation.

R= Correct, we have made these changes in the text. See lines 504-505, and lines 627-628.

Line 459. Change “reportedly” to “reported”.

R= done

Line 464-465. In “Our findings from the phylogenetic reconstruction based on nuclear markers revealed that these putative hybrids were intermixed between subclades of the three Bursera species”. Please, provide a reference supporting this statement, and explain what the expected topology would be when a hybrid zone occurs (Soltis et al. 2008. Systematic Botany 33(1):7-20; Okuyama et al., 2005. Molecular Biology and Evoluti 22(2):285-296). Same thing for lines 469-470 regarding haplotype networks; why is it expected that the haplotypes from putative hybrids will be intermediate links between parental species? What would be the difference between putative hybridization and the clinal variation mentioned in lines 456-457?

R= we removed that part of the text, as we eliminated the concatenated network. Regarding the intermixing of putative hybrids among species clades we added the reference from Week and Simpson 2007, and which find something similar in other Bursera species (see lines 531-535). For the study of Weeks and Tye (2009), where no evidence of hybridization was found, their species phylogeny did not showed major clades composed mainly with sequences of their presumed species, and instead samples were equally interchanged among species and putative hybrids. Moreover, the species identity was unclear. In our case, the taxonomic identity of the three species have been confirmed morphologically and genetically in phylogenetic studies (see lines 542-543). Our phylogenetic reconstructions showed that putative hybrids are the most intermixed within subclades of parental species, while the MJ networks showed that most frequent species haplotypes were shared in larger proportion with the putative hybrids. We have better specified along the discussion the evidence demonstrating patterns of hybridization. 

Lines 468. It is worth considering in the discussion that, given the lack of polymorphism in five regions of cpDNA, we cannot rule out that shared nuclear haplotypes between the three species and putative hybrids might be related to ancestral character retention or incomplete lineage sorting. In this sense, I recommend conducting recombination tests and also I recommend codifying genotypes for each individual which will allow conducting estimating allele frequencies, observed heterozygosity and test for admixture.

R= agreed, we have added the suggestion (see lines 571-575). Also, as suggested we conducted a recombination test and phased the sequences to calculate heterozygosity and admixture analysis. 

Line 510. Change “area” to “are”.

R= done

Line 582-589. Regarding the results for future projections of the species distribution models, what could be the potential implications of the predicted expansion or contraction of the hybrid zone -depending on the AOGCM- for the analyzed taxa?

R= We discussed in general what could occur in terms of hybridization, we have added a few more sentences regarding this point. See lines 666-673.

Reviewer #2: 

Title: Molecular evidence and ecological niche modeling reveal an extensive hybrid zone among three Bursera species (Section Bullockia) Manuscript ID: PONE-D-21-19713

Reviewer Summary: Quintero-Melecio et al. use a combination of niche modeling and population genetic analysis to assess and describe the presence of a hybrid zone among three species of Bursera in the section Bullockia. Their overall findings support the existence of an ecologically divergent set of hybrids with B. cuneata being the closest parental species to these hybrids. Niche modeling and patterns of niche divergence were further investigated to quantify the presence of the hybrid zone, including historical and future projections. These analyses supported a model of ecologically divergent hybrids, with an extensive current distribution.

Overall Impressions: Overall, I find the results in the manuscript to be compelling. The presentation is clear and the methodological choices are sensible, although they could use further justifications, assessment of assumptions, and descriptions in places (see below). WIth that said, I think some of the main conclusions could be bolstered in places. For example, is there no other possible putative hybrid occurrence point or points (from an herbarium or observational) to validate the extensive hybrid zone inferred from parental species overlap (as in Figure 4B)? Further comments are given below.

R= thanks Andrew for the useful comments to improve the quality of the manuscript, we appreciate your comments to better interpret our results. We have carefully addressed each of your comments, see below. 

Comments:

Pg. 4, lines 53-55. While hybrids are often phenotypically “intermediate” to “parent” species, they do not necessarily have to be so, which is consistent with some of your results from BAPS. I think there was a missed opportunity later in the manuscript to highlight the importance of genetic data, even from only two nuclear loci, to working out complex hybridization scenarios.

R= This is a very interesting comment, we agree with the reviewer in this point. We focus the identification of putative hybrids as morphologically intermediate individuals because this has been the traditional way in which Bursera hybrids have been identified in the field, and it was the easiest way of identifying them during our field sampling. However, we are aware that this may not be the case and, you point, this was evident from the results in BAPS. We have added this idea in the discussion and highlighted that hybridization in Bursera is far more complex than previously thought. See lines 561-568. 

Pgs 7 - 8, lines 127 - 150. How were heterozygous sites dealt with during sequence analysis (i.e., your sampled tree is a heterozygote for a polymorphic site)? For example, other studies have cloned PCR products for ETS in other Bursera species (see Weeks & Simpson 2004 Am. J. Bot. 91: 976-984). If they were ignored then would this not then create a bias? If these sites were called polymorphic then what was the procedure by which they were deemed true positives (e.g., comparing peak heights, areas, etc. on the chromatograms)?

R= this was a very important point that we missed in the beginning. As suggested by the reviewer 1 we have dealt with this issue by searching for unambiguous sites in the chromatograms and then phasing the sequences using PHASE (see lines 164-167). All posterior analysis were carried out with the phased sequences, which would eliminate the potential bias we have in the beginning. Overall, the results did not changed drastically, but now the data gives a better picture of hybridization. 

Pg. 10 line 197. What was the pairwise distance used in AMOVA? Was it simple p-distances or some transformation of those assuming a finite sites model? If p-distances, why not use pairwise distance from the best-fit model of sequence evolution from JMODELTEST?

R= The AMOVA was omitted in this version of the manuscript. We agree with your following comment, that given the nature of our data and ecological context, this analysis is not appropriate. 

Pg. 10, lines 193-200. Are the AMOVAs and Fst/Gst analyses really needed? My concern is that these methods, especially with complex hierarchies, are really designed for nested categories (i.e., trees nested in populations and populations nested in species), but with hybridization do you not have a different kind of categorical organization (something like a partially crossed design that violates the assumptions of the method where trees that are hybrids are really more like an interaction between two species rather than an independent category of the same meaning as the “parent” species categories)?

R= Agree. Thanks for pointing this issue, we have omitted this analysis in this new version. 

Pg. 11, lines 212-214. How were the thresholds of 20 km and 15 km arrived at for spatial thinning? I like that the threshold was lower for species with the smaller distributional area, but were there other criteria for choosing these values? Were the hybrids also thinned and, if so, why were they thinned?

R= We used different thinning thresholds for B. cuneata, because its distribution area is smaller relative to B. palmeri and B. bipinnata, this was the main criteria used. No spatial thinning was applied for the putative hybrids as the records were used only for the similarity test. Moreover, these records were not spatially overlapped (> 5km). We have made this clearer in the text; see lines 233-242. 

Pgs. 13 - 14, lines 273-275. Unequal sampling effort is definitely a problem given the variation in sample sizes for the “parent” species and the hybrids, but this is not the same as small sample sizes for the putative hybrids (n = 19). I think it would be helpful for the general reader to have a few arguments about why such small and spatially localized samples (see Fig. 4) are sufficient for this analysis.

R= We understand your point. Hybridization among Bursera species has been an overlooked phenomena by many botanists, and that is the reason there is such a low number of putative hybrid records. The small spatial extent of the used records (including were we sampled for genetic analysis) responds to the area that has been fully explored and were records are reliable (These observations were made by Jerzy Rzedowski, which is an authority in Bursera). We took a conservative approach by sampling reliable hybrid populations for genetic analysis (also due to financial constraints). For the ecological niche analysis, putative hybrid records were not used for estimating the overlap niche zone, but only the records of each parental species. In the discussion we have made clearer that our analyses refer to a potential hybrid zone, and that more hybrid populations across the potential hybrid zone are needed to better understand patterns of hybridization (see lines 586-591 and 643-645).

Pg. 14, lines 285-286. By equating the hybrid calibration area to the area of niche overlap of the “parent” species, are you not assuming that hybridization is successful anywhere and with any individuals from each species in this overlap area? Is this reasonable given that there are examples of variable hybridization success across ranges of species (e.g., Mandeville et al. 2017 Evol. Letters 1: 255-268) and that populations of trees often have strong patterns of local adaptation? Maybe some description and justification of this assumption is needed? This also applies to the measurement of the hybrid zone as given in Figure 4 and elsewhere because overlap of the “parent” species is being used as a measurement of hybrid presence.

R= As hybridization in Bursera is still a poorly known phenomena there are many unresolved questions, such as hybridization success and variation rates across the landscape. It is important to point that we are using the overlap zone only as a calibration area for hybrids PCA values in the similarity tests. This approach is appropriate in the M calibration area of Soberon concept, where is the area that encompasses the entire region that has been potentially colonizable for the lineage, in this case for the hybrid linage (Soberón, J., 2010. Ecography 33, 159–167; Barve et al. 2011. Ecological Modelling 11, 1810–1819.)

We have expanded the arguments in the discussion to point out that our model projections are a working hypothesis to evaluate in the future. As in other plant species, it is likely that hybridization varies considerably across the landscape given the local context of ecological and biological conditions. Hence, we have modified our interpretations about the predicted and potential extension of the hybridization zone to highlight that these results should be interpreted cautiously and should not be taken as unambiguous evidence of hybrid presence (lines 605-611). We have renamed this as potential hybrid zone across the text.

Pg. 13 lines 254 - 267. Was there a particular reason that ensemble models were not presented, maybe an ensemble across CCSM4.0 and MIROC5 for each RCP? This might help readers with the overall results from the ENMs.

R= MIROC5 and CCSM4 are scenarios constructed by different laboratories with different methodologies, so we decided to present the projections under these scenarios separately and not ensemble the models. 

Pg. 14, lines 296-297. For the niche equivalency and similarity tests using ecospat, did you specify the alternative flag in each function (i.e. -alternative = “greater” is niche conservatism and -alternative = “lower” is niche divergence)? If so, which direction did you specify and why? Lastly, why not use the niche overlap test?

R= We set “greater” in the similarity test as we wanted to test the conservatism between the niches between species. We have clarified this in the text. Also, we tested the niche overlap in terms of Shoener´s D and I metrics. See lines 295-298; 303-304.

Pg. 24, lines 512-516. It is not that you need markers with higher levels of polymorphism. It is that you would need many more unlinked markers, which is what SNPs derived from one of many different forms of GBS would do for you. Even then it may be difficult to parse hybrids into F1, F2, BC, etc. I think this sentence needs a slight tweak to emphasize the number of unlinked markers rather than “higher polymorphism”. Lastly, the point about sampling more populations of hybrids is valid, but is it not also sampling more of the geographical area of potential hybridization?

R= Correct, we have made this argument more precise as you suggest. See lines 586-591.

Editorial Suggestions:

Pg. 24, line 510. I believe “area” should be “are”.

R= true, corrected. 

It is my policy to sign all of my reviews: Andrew J. Eckert (aeckert2@vcu.edu)

R= We have formatted the manuscript following the journal requirements

2. Funding information should not appear in the Acknowledgments section or other areas of your manuscript. We will only publish funding information present in the Funding Statement section of the online submission form.

R= The funding was removed from acknowledgments.

3. We note that you have referenced (Rico et al. unpublished data) which has currently not yet been accepted for publication. Please remove this from your References.

R = It was removed from the ms. 

R= DNA sequences have been deposited in GENBANK.

---

## [Editor Report · Decision Letter 1]

1 Nov 2021

PONE-D-21-19713R1Molecular evidence and ecological niche modeling reveal an extensive hybrid zone among three Bursera species (Section Bullockia)PLOS ONE

Dear Dr. Rico,

Thank you for submitting your manuscript to PLOS ONE. After further consideration, we feel that it has been vastly improved after revision, but does not fully meet PLOS ONE’s publication criteria as it currently stands. Therefore, we invite you to submit a revised version of the manuscript.

The authors have done a solid job is responding to the reviewers' comments and criticisms. However, as I read through the revised MS, I found a large number of grammatical errors, misuses of commas and parentheses, and very long sentences that needed to be broken up. Since one of the requirements for publication in this journal is that all manuscripts be presented in proper English, I have returned the MS to the authors with edits so they can improve the presentation of their work for their readers. The paper is still quite long for the message, but I leave it to the authors to attempt to delete unneeded sentences and duplicate information. Please submit your revised manuscript by Dec 16 2021 11:59PM. If you will need more time than this to complete your revisions, please reply to this message or contact the journal office at plosone@plos.org. Please include the following items when submitting your revised manuscript:A rebuttal letter that responds to each point raised by the academic editor and reviewer(s). You should upload this letter as a separate file labeled 'Response to Reviewers'.A marked-up copy of your manuscript that highlights changes made to the original version. You should upload this as a separate file labeled 'Revised Manuscript with Track Changes'.An unmarked version of your revised paper without tracked changes. You should upload this as a separate file labeled 'Manuscript'.If applicable, we recommend that you deposit your laboratory protocols in protocols.io to enhance the reproducibility of your results. Protocols.io assigns your protocol its own identifier (DOI) so that it can be cited independently in the future. For instructions see: https://journals.plos.org/plosone/s/submission-guidelines#loc-laboratory-protocols. Additionally, PLOS ONE offers an option for publishing peer-reviewed Lab Protocol articles, which describe protocols hosted on protocols.io. Read more information on sharing protocols at https://plos.org/protocols?utm_medium=editorial-email&utm_source=authorletters&utm_campaign=protocols.

We look forward to receiving your revised manuscript.

Kind regards,

William J. Etges

Academic Editor

PLOS ONE
---

## [Author Response · Author response to Decision Letter 1]

4 Nov 2021

We are very thankful with the comments made by the academic Editor, William J. Etges, who made considerably edits of grammatical errors and other style issues. We have made all the suggested changes including broken up some large sentences. Moreover, we have considerably reduced the number of sentences, mostly for the discussion section where the manuscript was too long. The sentences included unnecessary information or duplicated information with introduction and discussion. A detailed record of the changes made can be revised in the manuscript with track changes.

---

## [Editor Report · Decision Letter 2]

9 Nov 2021

Molecular evidence and ecological niche modeling reveal an extensive hybrid zone among three Bursera species (Section Bullockia)

PONE-D-21-19713R2

Dear Dr. Rico,

We’re pleased to inform you that your manuscript has been judged scientifically suitable for publication and will be formally accepted for publication once it meets all outstanding technical requirements.

Kind regards,

William J. Etges

Academic Editor

PLOS ONE
---

## [Editor Report · Acceptance letter]

11 Nov 2021

PONE-D-21-19713R2 

Molecular evidence and ecological niche modeling reveal an extensive hybrid zone among three *Bursera* species (Section *Bullockia*) 

Dear Dr. Rico:

I'm pleased to inform you that your manuscript has been deemed suitable for publication in PLOS ONE. Congratulations! Your manuscript is now with our production department. 

Kind regards, 

on behalf of

Dr. William J. Etges 

Academic Editor

PLOS ONE